# Crystal dissolution by particle detachment

Guomin Zhu[1,2], Benjamin A. Legg[1], Michel Sassi[1], Xinran Liang[1], Meirong Zong[1], Kevin M. Rosso [1] & James J. De Yoreo [1,2] ✉

Crystal dissolution, which is a fundamental process in both natural and technological settings, has been predominately viewed as a process of ion-by-ion detachment into a surrounding solvent. Here we report a mechanism of dissolution by particle detachment (DPD) that dominates in mesocrystals formed via crystallization by particle attachment (CPA). Using liquid phase electron microscopy to directly observe dissolution of hematite crystals – both compact rhombohedra and mesocrystals of coaligned nanoparticles – we find that the mesocrystals evolve into branched structures, which disintegrate as individual sub-particles detach. The resulting dissolution rates far exceed those for equivalent masses of compact single crystals. Applying a numerical generalization of the Gibbs-Thomson effect, we show that the physical drivers of DPD are curvature and strain inherently tied to the original CPA process. Based on the generality of the model, we anticipate that DPD is widespread for both natural minerals and synthetic crystals formed via CPA.

Dissolution has been extensively studied due to its importance in mineral weathering[1], materials degradation and corrosion[2,3], drug release[4,5], and crystallization[6,7]. Rates of dissolution are in part material specific and are determined by composition[7,8], defect content[9–11], and the extent of departure from equilibrium[10,12–14]. The main mechanistic model for understanding most dissolution behavior has been based on a picture of monomer-by-monomer detachment events (e.g., atom-by-atom or ion-by-ion) with local structure defined within the terrace-ledge-kink framework of crystal growth[6,7]. Essentially the inverse of crystal growth[15], this picture has been used to understand most dissolution phenomena as resulting from ion removal at specific sites, such as step edges[10], often in etch pits[10,13], as well as along strained bulk defects such as dislocations or grain boundaries[11,14].

Laboratory studies of dissolution kinetics in many fields[2–5,16,17], which are typically performed by monitoring rates of material release into solution or loss from material surfaces, have long pointed to the importance of quantifying the evolution of reactive surface area. Doing so provides a basis for normalizing overall rates measured over time at steady state and then associating them with what are presumably constant underlying rates of monomer detachment. Different dissolution kinetics models are typically then fit to the data to understand the dissolution behavior[18]. Except for limited cases where the micro to nanoscale topology can be monitored during dissolution using techniques like in-situ liquid phase AFM[15,19], the evolution of topology is typically not directly accessible during experiment. The kinetics is further confounded by the fact that local interfacial curvature can alter dissolution rates through the effect of Laplace pressure on solubility, as described by the Gibbs-Thomson equation. This effect is easily modeled for simple geometries (such as spherical particles), where it makes well-known predictions about the enhanced solubility of small particles, However, the dissolution of hierarchically structured materials is more difficult to treat. Thus, the combined influence of reactive surface area and particle geometry have historically been difficult to disentangle.

In recent years, a paradigm of crystal growth has emerged, wherein classically assumed processes of atom-by-atom or ion-by-ion addition have been supplemented by mechanisms of crystallization by particle attachment (CPA)[20–22]. CPA, particularly a sub-class known as oriented attachment occurs when aggregating particles either fuse to form single crystals, or generate assemblies of aligned but distinct particles known as mesocrystals in which the particles are separated by mostly organics[23] or hydration layers[22]. Because the crystallinity of the primary particles biases the modes of attachment, the resulting structures are often distinct from those of traditional aggregates, such as the fractal aggregates that are commonly formed by diffusion limited and reaction limed aggregation[24]. Examples include porous hematite spindles[25,26], one dimensional chains and two dimensional superlattices of semiconductor nanoparticles[27,28], branched chains of

[1]Physical Sciences Division, Pacific Northwest National Laboratory, Richland, WA 99354, USA. [2]Department of Materials Science and Engineering, University of Washington, Seattle, WA 98195, USA. ✉e-mail: james.deyoreo@pnnl.gov

anatase particles[29] and boehmite particles[30], goethite nanorods with extensive internal defects[31], and even assemblies of peptides[32] and proteins[33].

Although research on CPA has drawn significant interest in recent years[20], whether, by analogy, the reverse process of dissolution by particle detachment (DPD) occurs and what determines its contribution to the overall dissolution rate relative to concomitant classical monomer-by-monomer dissolution has received little attention. One rare example is the "chunk effect" that has been observed in electrochemical corrosion[3,34–37], wherein the disintegration of polycrystalline materials into individual particles is believed to explain anomalously high dissolution rates. However, the polycrystals do not arise from CPA and an electric field is usually involved to generate the effect. None the less, past observations that OA-generated mesocrystals in which the primary particles are separated by intervening hydration layers can disintegrate into the primary particles during sublimation of the hydration layers[22] provides evidence that mesocrystalline solids grown via CPA can undergo DPD. However, whether or not a similar process is possible when the constituent particles have fused to form a continuous crystal lattice has not been explored. Understanding the mechanisms of DPD will be important to modeling the durability and reactivity of diverse materials, especially for the complex hierarchically structured materials that are frequently synthesized using CPA.

## Results

### Direct observation of dissolution

To investigate the existence and mechanisms of DPD, we synthesized hematite (Hm, $Fe_2O_3$) particles with two different morphologies: compact rhombohedra (rhHm) (Fig. 1a–d), and spindle-shaped mesocrystals (spHm) having a branched, porous structure constructed from spheroidal primary particles (Fig. 1e–h). Electron diffraction patterns for rhHm (Fig. 1b) show a sharp, monocrystalline pattern (Fig. 1b inset) that indicates good crystallinity, as is further verified by high resolution transmission electron microscopy (HRTEM) (Fig. 1c) and scanning TEM (STEM) (Fig. 1d). In contrast, the spHm mesocrystal diffraction patterns along the same zone axis as for rhHm (Fig. 1f) are elongated, reflecting slight misorientations of the primary particles. HRTEM (Fig. 1g) and HRSTEM (Fig. 1h) images show a continuous lattice across the particles, but with slight misorientations between them, often associated with line defects (Fig. 1h).

Many methods have been used to explore dissolution kinetics and pathways, but very few techniques can directly image the dissolution process at a nanometer scale in real time. In-situ liquid phase (S)TEM, is an emerging technique that has shown tremendous progress in revealing pathways and exploring the kinetics of nucleation[38], crystal growth[39,40], assembly[41], and dissolution[42,43]. Using in-situ liquid phase (S)TEM, we directly observed the dissolution of those two types of Hm structures. The results show that the rhHm exhibits typical single crystal dissolution behavior in which the corners and edges dissolve faster than the faces and the particles become rounded over time, eventually disappearing (Fig. 1i and Supplementary Movies 1, 2, and 3). Similar results have been obtained previously[44,45]. In comparison, the spHm mesocrystals exhibit a more complex morphological evolution. Initial dissolution reveals a branched skeleton backbone, consisting of rod-like chains of particles (Fig. 1j, k, Supplementary Movie 4), which can reasonably be assumed to be the remnants of the primary particles originally forming the mesocrystal, based on both the ex-situ TEM images (Fig. 1e–h) and our previous in-situ TEM results documenting their formation[26]. In addition. the center portion of the spHm, which is highlighted by the grey particles (Fig. 1k), dissolves faster than the rest (Supplementary Fig. 1c). We hypothesize that this anisotropic dissolution, which is found to be universal across all the spindles independent of orientation (Supplementary Fig. 1a), is related to the underlying microstructure determined during spHm formation.

### Dynamics of necking before particle detachment

We further applied in-situ liquid phase HRTEM to capture the details of the spHm dissolution process. The results show dissolution involves the formation of necks in thin regions of the branches followed by particle detachment and rapid dissolution of the emitted particles. The dense structure of the spHm at their early stages makes it difficult to image the detachment details, but this process can be clearly observed during later stages, when only the branched skeleton remains. Time dependent HRTEM images (Fig. 2a, b, Supplementary Fig. 2, and Supplementary Movies 5, 6, and 7) directly reveal neck formation and subsequent break-up, which allows individual particles to separate from the branched skeleton after which they can move freely. An example trajectory of a detached particle is shown in Supplementary Fig. 3. This process of DPD is the compliment of the inverse process of CPA.

Where necks form, we observe that they dissolve more rapidly than the body (Fig. 2d). While the body of each particle dissolves at a more or less constant rate (Fig. 2d, open symbols), the necks shrink at an accelerating rate with time (Fig. 2d, closed symbols). Moreover, upon shrinking to a diameter of ≈1.0 nm, the necks disintegrate, and the particles suddenly detach, thus the neck size jumps to zero (Fig. 2d). At diameters of under 1 nm, the neck only contains a few tens of atoms, and likely becomes difficult to sustain. Nonetheless, we estimate the surface free energy gain to be around 1273 meV during the neck breakup, which is equivalent to changing the neck size from 1 nm to zero, using an interfacial energy of 0.13 J m$^{-2}$ for hematite[46]. This suggest that thermal energy itself, which is only about 25 meV cannot drive neck disintegration.

More details of this neck break-up will be discussed later. A histogram of neck-sizes observed immediately prior to detachment (Fig. 2e) shows values of 1.0 ± 0.1 nm. This value also corresponds to the lower bound of the neck sizes measured in ex-situ HRTEM images (Fig. 2e), confirming that necks smaller than this are not stable.

We utilized various TEM imaging modes and electron beam (e-beam) dose rates to explore the influence of the e-beam on dissolution of spHm. In STEM mode, we also observe necking and particle detachment (Supplementary Movie 8 and Supplementary Fig. 4), though the movement of detached particles is too fast to track. During TEM imaging, we find that higher dose rates lead to higher dissolution rates (Fig. 2f, Supplementary Movies 9 and 10). To exclude the influence of direct e-beam sample interactions on dissolution, we manipulated the beam position (Supplementary Fig. 5) to selectively irradiate only half of a Hm spindle and compared the dissolution behaviors of the halves. We find that the spindles dissolve in the region with no beam irradiation, which implicates the role of diffusion of radicals in the dissolution. In another study[47], $e_{aq}^-$ and $O_2^-$ were found to be key radiolytic species contributing to akageneite (β–FeOOH) dissolution under the e-beam. If we assume that those species also play a role in Hm dissolution, we then estimate the hematite surface accessible to these beam-generated radicals extends beyond the irradiated region by the characteristic diffusion length of $e_{aq}^-$ and $O_2^-$. At a pH of 7, for the high dose rate, the characteristic diffusion length of $e_{aq}^-$ and $O_2^-$ length are 130 nm and 200 nm, respectively, while for the low dose rate, they are 160 nm and 250 nm, respectively. The diffusion length here is calculated by taking the square root of the average lifetime of the radicals multiplied by the diffusion coefficient. (Supplementary Table 2 and Supplementary Figure 6).

### Atomic structure of the neck and analytic model

To gain atomic-level structural information about the dissolving spHm, we applied ex situ TEM to partially dissolved spHm (see details in Methods). The images reveal a time dependent structural evolution from spHm to a branched skeleton with necking similar to that observed in situ (Fig. 3a–d and Supplementary Fig. 7). Based on the progression of dissolution for the spHm, we can illustrate the spindle

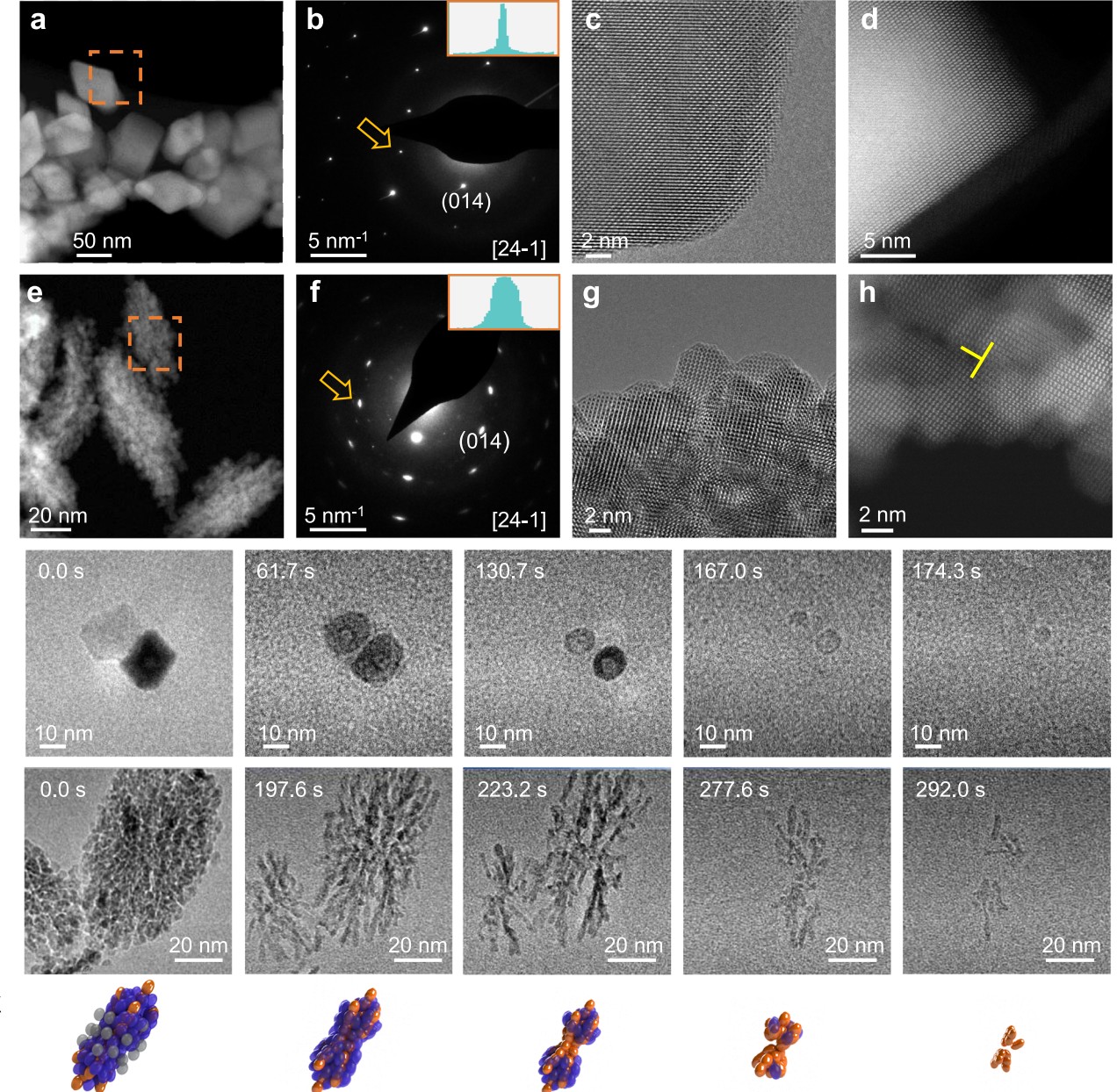

**Fig. 1 | Morphological difference between rhHm and spHm mesocrystals and the distinct dissolution behavior of each. a, e** Low magnification STEM image of rhHm and spHm. **b, f** Corresponding electron diffraction of rhHm and spHm along the same zone axis in **a** and **e**, taken from areas highlighted by the yellow dashed square. The insert is the intensity profile drawn across the diffraction spot indicated by the orange arrow. **c, g** TEM image of rhHm and spHm. **d, h** STEM image of rhHm and spHm. The yellow marker indicates an edge dislocation in **h**. Comparison of **a–d** and **e–h** shows that rhHm has a higher crystallinity with a smoother surface than spHm, which consists of many primary particles, between which there is often a lattice mismatch. **i** TEM image series showing dissolution of rhHm particles. **j** TEM image series showing dissolution of spHm. **k** Cartoons showing the dissolution of spHm. Grey particles dissolve the fastest, then blue particles, and, finally, the orange particles dissolve the slowest.

structure with a half-spindle model showing the hierarchy of dissolution inside the spindle that leads to the skeleton of rod-like structures (Fig. 3h). The particles in grey dissolve first, followed by the particles in blue, revealing the skeleton in brown, which dissolves last. HRTEM images of these structures also provide atomic details at the necks that show there is a range of orientations between the primary particles, with the mismatch between adjacent particles ranging from about 0° to 10° for the cases investigated here (Fig. 3e–g). As mentioned previously, the continuous nature of the crystal lattice is apparent; this misorientation between grains appears to be facilitated by dislocations, as for a low-angle grain boundary (Fig. 3f). The grain boundary–

i.e., strain–energy at these contacts can be estimated from the Shockley-Read equation, which dictates that it increases linearly with respect to the degree of grain mismatch below a certain angle[48]. The greater recalcitrance to dissolution by the interior skeleton suggests that this region has coarsened to reduce the defect density, coarsen the particles, and eliminate concavities in the branches.

To understand the roles of particle misorientation and interfacial curvature, we developed a general model for dissolution that accounts for the impact of surface energy, $\gamma_s$, grain boundary energy, $\gamma_{GB}$, and particle shape. This model (explained in detail in the Supplemental Materials) considers how interfaces (including the particle surface

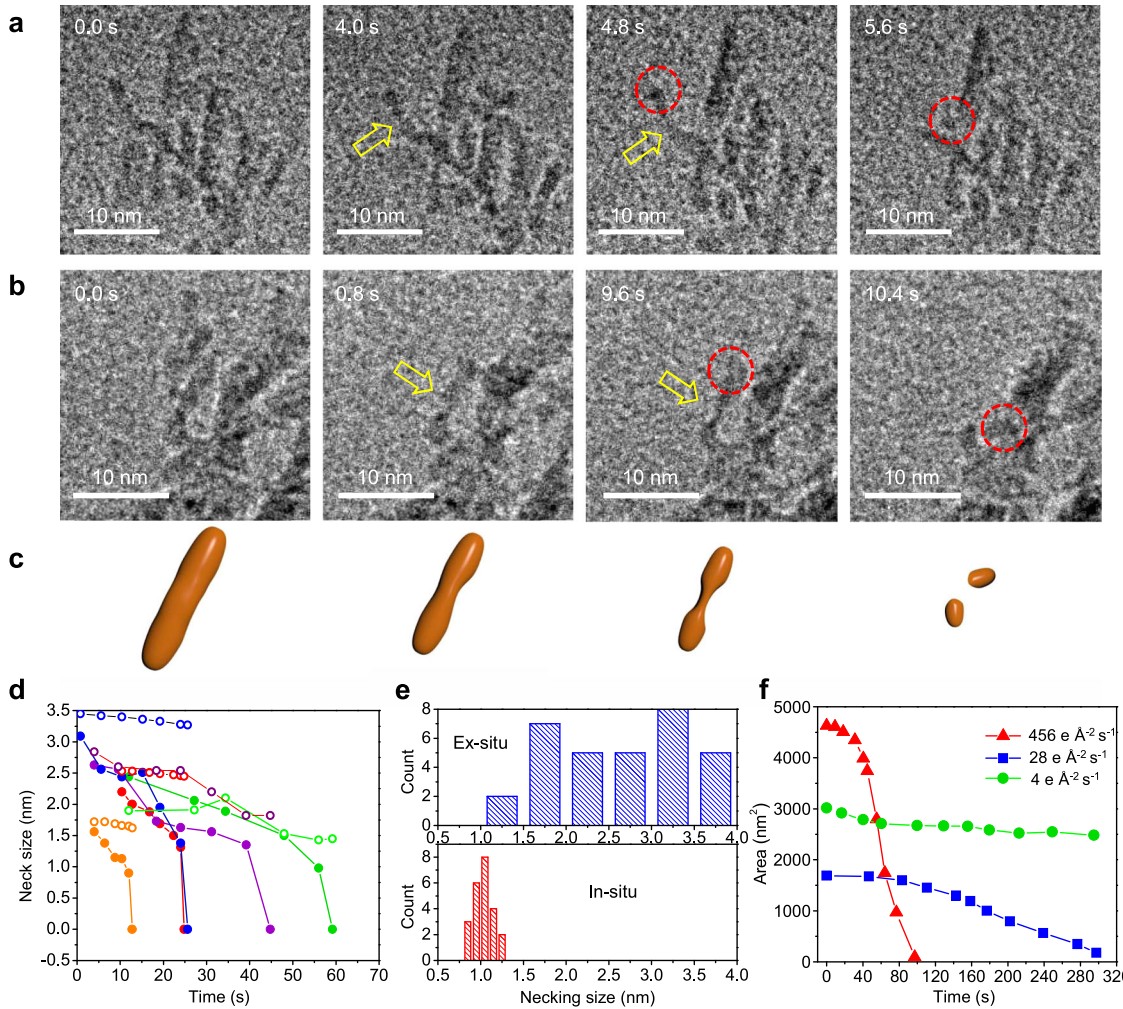

**Fig. 2 | Necking and particle detachment during spHm mesocrystal dissolution.**
**a**, **b** Selected time-lapse TEM images showing the necking process (yellow arrows) and eventual particle detachment (dashed red circles), extracted from Supplementary Movie 5 and 6, respectively. **c** Schematic of necking and particle detachment process. **d** The dissolution kinetics for the necks (solid symbol) and particle bodies close to the necks (open symbol) showing the abrupt drop to zero at a neck size ≈ 1 nm. The colors represent five necking events from Supplementary Movies 5 and 6. **e** Distribution of neck diameters from ex-situ TEM images (top) and minimum neck diameter preceding particle detachment from in-situ TEM experiments. **f** Dose rate dependent dissolution kinetics of spHm. The colors represent three dose rates.

area, $S_s$, and grain boundary area, $S_{GB}$) produce a Laplace pressure that changes the thermodynamic driving force for dissolution. Our model calculates the Laplace pressure in terms of generalized coordinates, and thus allows us to easily treat the dissolution of arbitrary shapes. Here, we model necked particles as a series of conical segments, defined by the radii, $r_i$, where each segment meets. In brief summary, the Laplace pressure associated with a coordinate, $q_i$, is obtained from the ratio of the derivative in interfacial energy with respect to that coordinate ($\gamma \partial S/\partial q_i$) to the derivative in volume ($\partial V/\partial q_i$). The Gibbs-Thomson equation then allows us to convert the Laplace pressure into changes in local solubility about each segment. Increased solubilities are modeled as an enhancement in the local ion detachment rate (per unit area), which provides the basis for a kinetic model that predicts the evolution of particle shape.

The solubilities predicted by our model agree with standard implementations of the Gibbs-Thomson equation, which calculate the Laplace pressure in terms of local Gaussian curvature. Simple spherical and cylindrical particles have a positive surface curvature that leads to enhanced solubility for smaller particles. Thus, smaller spherical particles dissolve at an enhanced rate (Fig. 4a, b). However, the necks between particles display a combination of positive curvature (which favors deeper incision at the neck) and negative curvature (which favors neck healing). Competition between these effects determines the evolution of neck shape.

Our models predict that the negative curvature dominates for sharp necks, which typically cause the neck to heal. Thus, most initially sharp necks will quickly disappear and not contribute significantly to particle detachment. (Fig. 3i, l). For wider necks, the positive curvature becomes dominant. These necks exhibit some initial healing, but ultimately trend towards more rapid dissolution at the neck than at the adjacent branch, leading to detachment (Fig. 3j, l). However, the neck profile always remains wide and gently curving, and by the time a neck finally separates, the rest of the branch has significantly dissolved too. Thus, this pathway is also unlikely to produce long-lived individual particles. A third scenario is given by a branch with no initial neck but with a high energy grain boundary (Fig. 3k). The grain boundary also contributes to the Laplace pressure, and causes a sharp incision to form that leads to rapid neck dissolution and break-up (Fig. 3k, l).

Thus, these three scenarios exhibit distinct dissolution dynamics (Fig. 3l). More complex dynamics could be expected in a real neck configuration where curvature is variable and is accompanied by varying levels of grain boundary energy. These three scenarios and their combinations provide a rationale for why only certain regions of neck break-up are observed in situ. The TEM results (Fig. 2a, b) could

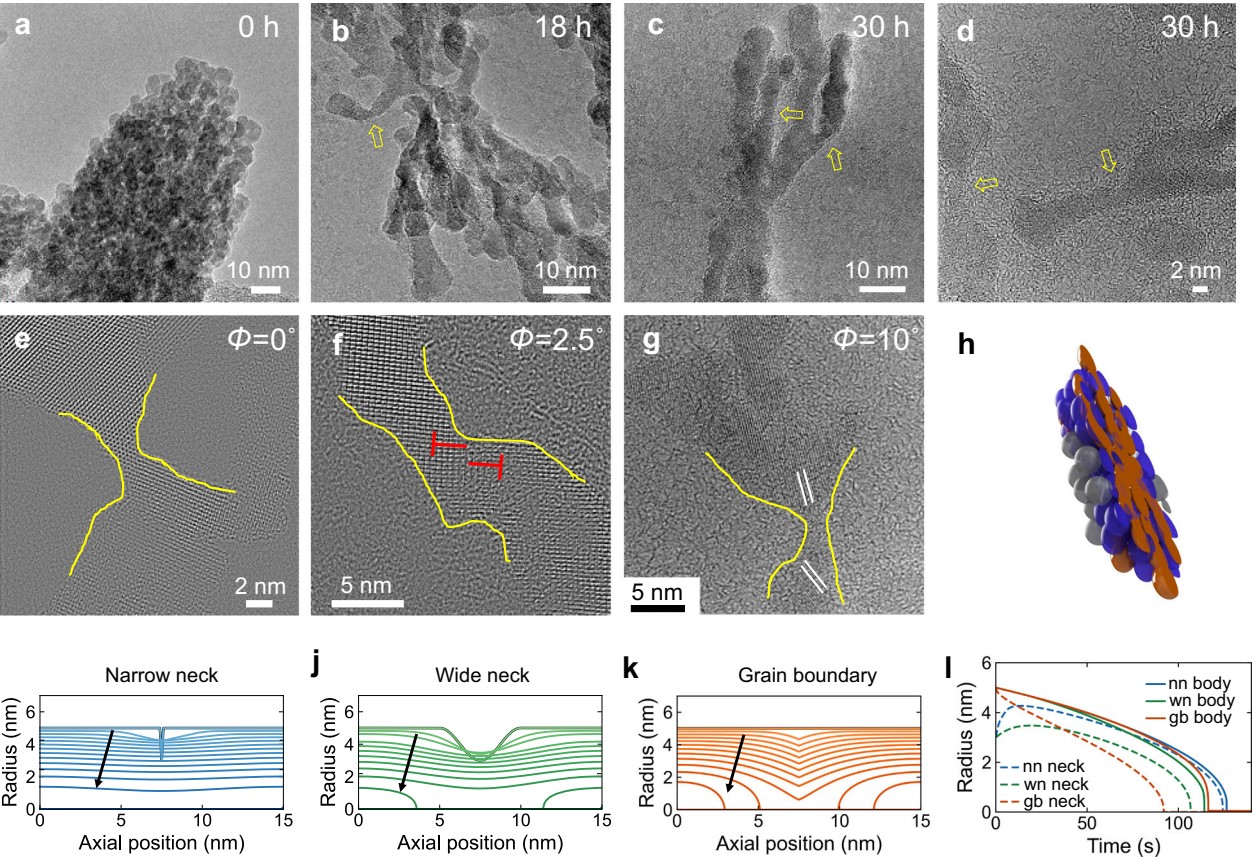

**Fig. 3 | Atomic structure of spHm necks and simulations of the influence of geometry and grain boundary energy on neck dissolution. a–d** Ex-situ TEM images of the spHm dissolution over time. A skeleton of rod-like structures is revealed during the dissolution. Necks are highlighted by the arrows. **e–g** HRTEM images of necks showing the angular shift Φ in the projected crystallographic orientation across adjacent particle domains. **h** Schematic of spindle structure illustrating the hierarchy of reactivity, which decreases in the sequence of grey to blue to brown. **i** Dissolution profile of a narrow neck (nn) showing some initial healing followed by dissolution at a rate similar to that of the adjacent body. **j** Dissolution profile of a wide neck (wn), which leads to break-up of the neck. **k** Strong influence of strain energy on neck evolution, showing rapid neck dissolution and breakup. The contours in **i–k** indicate the surface profile at evenly spaced increments of time advancing in the direction of the arrows. **l** Dissolution dynamics of the neck and body in **i**, **j** and **k**.

be explained by a branch that contains no initial neck in the beginning (0.0 s), but where a high-energy grain boundary causes the necking to initiate, and gradually form a wider and deeper neck (Fig. 2a, 4.8 s and Fig. 2b, 9.6 s, indicated by the arrow). Eventually, the rapid dissolution at the neck causes the particles to break-up and detach.

**Quantitative analyses of the dissolution**

To quantify the dissolution rate at the single particle level, we start by analyzing the time dependence of rhHm particle size, which is fit well by the Gibbs-Thomson equation (Fig. 4a, Supplementary Movie 1), thus showing that smaller particles dissolve faster. The spHm consists of primary particles that, individually, are smaller than the rhHm. More importantly, as shown above, necking and particle detachment occur during dissolution, so we expect spHm to dissolve faster than rhHm. To directly compare the rates, we mixed rhHm and spHm to ensure both are subject to same environment (Fig. 4b, see details in Methods). The time dependence of the rhHm dissolution rate is again well described by the Gibbs-Thomson relationship (Fig. 4b, blue and green curves). Analyzing the dissolution data for spHm is not as straight-forward as the outer particles first dissolve to form the skeleton of rod-like chains of particles before fully dissolving. However, the spindles are almost completely dissolved after ≈119 s. In comparison, the lifetime of a 6 nm spherical particle (which is close to the size of the primary particles that comprise the spHm) is predicted to be ≈90 s (Fig. 4b, orange curve). Thus, the timescale of dissolution for the

spindles appears to be defined by the primary particle size, rather than the aggregate size.

To compare the macroscopic dissolution rates of spHm and rhHm, we measured their temporal dissolution profiles using inductively coupled plasma-optical emission spectrometry (ICP-OES) (Fig. 4c). The dissolution of spHm and rhHm are both fast in the beginning and then level off, but the dissolution of spHm is much faster (≈7.5 times) than that of rhHm, in agreement with the faster dissolution rate for spHm versus rhHm at the single particle level. The reduction in dissolution rate with time for both cases can be explained by the reduction in surface area during dissolution. The mass dissolved rate, $dM/dt$, is given by $4\pi\rho r^2 dr/dt$, in which $\rho$ is the density, $M$ is the dissolved mass at $dt$, and $r$ is the radius of the particles. The magnitude of $dr/dt$ should be proportional to $1/r$ due to the Gibbs-Thomson effect, so the dissolution rate $dM/dt$ is proportional to $r$, and therefore the dissolution gets slower.

We want to emphasize that the enhanced dissolution rate of the spHm over that of the rhHm cannot be obtained simply by assuming the observed rate for rhHm and adjusting for surface area. Because of the high curvature of the constituent particles, the necks between particles, and the particles that detach, the measured rates are three times greater than expected based on increased surface area alone. While this enhancement is a clear implication of Fig. 4b, the comparison of macroscopic dissolution rates demonstrates this phenomenon at the macroscopic level.

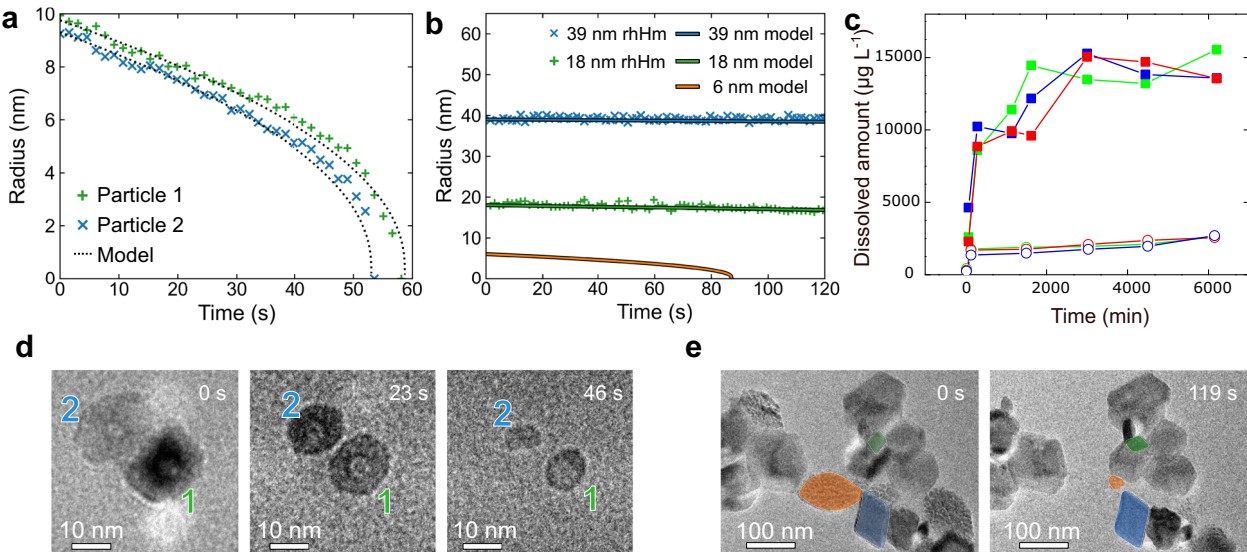

**Fig. 4 | Quantitative analyses of the dissolution based on the Gibbs-Thomson effect. a** Dissolution kinetics of the rhHm fitted by Gibbs-Thomson equation and associated snapshots of the dissolution (**d**) (Supplementary Movie 1). **b** Use of the Gibbs-Thomson equation model of different sizes to understand the dissolution of rhHm and spHm. The color in the plot corresponds to the particles highlighted in the images in **e** (Supplementary Movie 11). Notice that the orange plot corresponds to the 6 nm model. The radius of the spindle is not plotted as the spindle has a complex morphological evolution. **c** Total amount of dissolved Fe versus time for rhHm (open symbols) and spHm (closed symbols) as determined by ICP-OES. The colors represent separate tests on the same condition.

The above findings reveal the previously unknown dissolution pathway of DPD, which accelerates dissolution due to the small size of the constituent particles and the strain produced at the interparticle boundaries by the CPA process, which is recognized to occur widely in materials systems as diverse as semiconductors[27,28,49,50], metals[39,40], silicates[51,52], oxides[21,22,26,31,53–55], fluorides[56], carbonates[54], organic compounds[57], peptides[32], and proteins[33]. Hence, even though classical monomer-by-monomer dissolution will always be present and, indeed, is still the fundamental process that enables DPD to take place, the nanoscale structure of mesocrystals in this wide range of materials fundamentally alters their dissolution dynamics from that of monolithic single crystals and is thus a direct consequence of growth via CPA. We fully expect these dramatic differences in dissolution behavior to be manifest in other properties. For example, a particle aggregate comprising a dendritic arrangement of rod-like particle chains can be expected to deform through bending of the arms and slip at the interparticle boundaries, possibly leading to a stress-strain relationship resembling plastic deformation instead of that for a brittle single crystal.

The extent to which rates of DPD and resultant dissolution morphologies differ from those seen in compact crystals grown by the classical process of ions or ion clusters addition to steps on faceted crystal surfaces depends on inherent materials parameters, such as solubility, surface tension, and elastic constants (via the grain boundary energy), as well as external factors of temperature and incubation time. While the expressions used in the model presented here are general and include the inherent materials parameters explicitly, the external factors are only implicitly incorporated into the analysis through the as-observed morphology. Higher temperatures and/or longer times spent in solution near the saturation point lead to more extensive coarsening, which increases average particle size, reduces curvature, relieves misalignment and corresponding strain at the boundaries, and eliminates concave regions[26,58,59] to an extent that also depends on the atomic mobility of the material. The implication is that the extent to which DPD dominates the dissolution process for a given material system cannot be predicted simply based on materials parameters alone; rather, either priori knowledge of the CPA dynamics that led to formation of the mesocrystals or observational data on the resulting morphology and distribution of grain boundaries is required. Conversely, the influence of both internal and external factors implies an ability to tune the speed of the dissolution using methods to control the primary particle size and strain, which could be applied in areas like corrosion engineering, nutrient release, and drug delivery. Future work should explore the factors that define the grain-boundary energies between particles in mesocrystals and how this influences the degree to which DPD occurs.

Potential applications that are impacted by either the realization that DPD can dominate dissolution processes or by the opportunities that DPD offers to manipulate dissolution processes when materials are formed via CPA fall into two categories: 1) using the phenomenon of DPD to interpret observations in geological settings and materials processing systems and 2) utilizing the phenomenon to tune dissolution rates in technological settings. In the first case, understanding that minerals formed by CPA—and, thus, those that exhibit a hierarchical morphology—will dissolve at a higher rate, release nanoparticles as they dissolve, and do so in a way that preferentially maintains regions that have undergone the most coarsening, will be critical to interpreting downstream outcomes. Examples include rates and resulting microstructures associated with mineral weathering and/or recrystallization in geochemical reservoirs, riparian environments, and in industrial settings. The latter include processing of aluminum ores into suspensions of aluminum hydroxide particles during the Bayer process[60] and the processing of legacy nuclear wastes, which are characterized by extensive aggregation of primary nanoparticles[30,61]. In the second case, as discussed above, creating conditions in which target products form via CPA and using methods to control the primary particle size and strain would establish an ability to tune the rates and products of dissolution, nanoparticle release, and recrystallization with applications in areas like corrosion engineering, nutrient release, and drug delivery.

In conclusion, growth of crystals by CPA has consequences in materials synthesis where superlattice formation and particle branching produce classes of materials not accessible through classical growth, leading to additional optical, electronic, and catalytic

properties. CPA also has consequences in geologic settings where growth rates, isotopic signatures, impurity concentrations, and microstructure will all be changed from that which arises through classical ion attachment at step edges. Here we add to the list of differences, the dissolution behavior of crystals generated through CPA from that of monolithic crystals, with consequences for a range of phenomena, such as mineral weathering, reactive transport, recrystallization, release of nutrients, and drug delivery. The analyses presented here provide a starting point for predicting how DPD influences such phenomena. The exact role of DPD needs to be carefully studied to understand how it both depends on and impacts microstructural evolution and to connect its characteristics to the above phenomena.

## Methods

### Materials
Sodium oxalate (99.99%, 379735-5 G), ferric nitrate nonahydrate (99.95%, 254223-10 G), and sodium hydroxide (99.99%, 306576-25 G) were all purchased from Sigma-Aldrich. Nuclease-free water (438736-1 L) was purchased from ThermoFisher Scientific.

### Synthesis of spHm
Spindle Hm (spHm) was synthesized following a procedure adapted from Schwertmann and Fischer[62-64]. Briefly, 2-line ferrihydrite (Fh) was synthesized by mixing 0.6 mL iron nitrate (72 mM) and 0.40 mL sodium hydroxide (0.32 M) in a 15 mL Teflon container. 15 μL sodium oxalate (0.253 M) was added to the mixed solution, and the final oxalate concentration was kept at about 2 mM. The final pH of the mixed solution was adjusted by adding sodium hydroxide solution and nitric acid solution and kept in the range of 6.0–8.0. The container was incubated in an oven at 90 °C without any light illumination. We scaled up the synthesis to perform BET measurement and ICP measurement.

### Synthesis of rhHm
The growth of rhombohedral Hm (rhHm)[64,65] was achieved by mixing 0.6 mL iron nitrate (72 mM) and 0.38 mL sodium hydroxide (0.32 M) solution, without oxalate in a 15 mL Teflon container. The pH was adjusted to neutral using sodium hydroxide or nitric acid solution. The solution was then heated in the oven at 90 °C for 10 h.

### ICP measurement of the dissolution kinetics
We first mixed 4.5 mL spHm solution, 3.5 mL oxalic acid (0.26 M), and 72 mL $H_2O$. The pH is about 1.9, which is close to the calculated pH (2.0) from the oxalic acid concentration. Three Parallel studies were carried out applying the same solution and concentration. We drew 1.5 mL solution and centrifuged the solution at 10000 rpm, with a centrifugal force of 7828 × $g$, for 5 min, then took out 1 mL supernatant and applied a filter with 100 nm sized pore (Durapore PVDF membrane from MilliporeSigma) to filter the solution twice for the supernatant. The first three drops of the filtered solution were discarded. The solution was then diluted accordingly by adding 2 wt.% $HNO_3$. All solutions were split for analysis in the laboratory. The pseudo total amounts of Fe in solution samples were determined by inductively coupled plasma atomic emission spectroscopy (ICP-OES; Perkin Elmer Optima 3000) at PNNL. For ICP-OES analysis, the samples were passed through a 0.45 μm nylon filter, then were diluted using 2 wt.% $HNO_3$ to give a final concentration lower than 200 mg $L^{-1}$. All reagents were of analytical grade. Blank samples with reagents yielded negligible results. For QA/QC, the standard NIST 2710 was analyzed together with samples. The detection limits of ICP-OES were 0.05 mg $L^{-1}$ for Fe.

### Ex-situ dissolution TEM experiments
We took about 1 μL solution from the dissolution experiments every several hours, cast drop it on the TEM grids, and air dried it. For general characterization, regular lacy TEM grids were used, while for high-resolution TEM imaging, graphene TEM grids were used to minimize the background noise and stabilize the sample simultaneously. For the Hm close to the complete dissolution, we took out about 1 mL solution and centrifuged it at 12,000 rpm, with a centrifugal force of 9394 × $g$, for about 2 min to concentrate the sample before making the TEM sample.

### In-situ liquid phase TEM/ STEM experiments on dissolution
In a typical experiment, 0.4 μL spindle suspension was dropped on a 30 nm thick $SiN_x$ spacer chip with a spacer of 100 nm. It was then quickly covered by another 30 nm thick $SiN_x$ window chip to minimize liquid evaporation. Both windows were plasma cleaned for 1 min at medium power before the $SiN_x$ assembly. We cleaned the tip of the holder (Hummingbird Scientific), tweezer, and glass slide by sonicating them in water for several minutes and then drying them afterward by applying compressed nitrogen. We wore masks during the whole assembly procedure to minimize potential organic contamination from the breath. A 200-kV FEI Tecnai $G^2$ microscopy was used with a 2048 × 2048 pixel charge-coupled device camera (FEI). TEM is operated at 3900 extracting voltage, with an acceleration voltage of 200 kV, the gun lens is 6. During the in-situ experiment, $SiN_x$ chips act as a container that encloses the solution, which contains Fh. To exclude the influence of the container on the Hm dissolution, reference experiments were performed by adding the liquid cell chips into the solution for the dissolution. For the spHm dissolution, we mixed 100 μL spHm suspension synthesized, 800 μL $H_2O$, and 100 μL $HNO_3$ (3 mM). The final pH of the mixed suspension is around 3.5. For the rhHm dissolution, we replaced the spHm suspension with the rhHm suspension. For the in-situ dissolution experiment of rhHm and spHm at the same time. We mixed 400 μL hematite suspension and 400 μL rhHm suspension, together with 800 μL $H_2O$ and 100 μL $HNO_3$ (3 mM). We sonicated all the suspension for 2 min and then drop the mixed suspension onto the spacer chip.

### BET surface measurement
The specific surface area of air-dried $Fe_2O_3$ samples including the pore size distribution, was determined by BET (Brunauer, Emmett and Teller) $N_2$ gas adsorption and it was measured with particle analysis (Micromeritics ASAP™ 2020) instrument. Prior to analysis, the samples were degassed overnight at 50 °C under vacuum. The surface measurement for the spHm is 87.5 $m^2 g^{-1}$, and it is 34.3 $m^2 g^{-1}$ for the rhHm.

### Dose rate for the radical calculation
The dose rate produced by a 300 kV electron beam has been calculated using the formula[66].

$$\psi = \frac{10^5 S_p I}{\pi a^2} \quad (1)$$

Where $\psi$ is the dose rate (in Gy $s^{-1}$), $10^5$ is a conversion unit factor, $S_p$ is the stopping power of an electron in liquid water (in MeV $cm^2 g^{-1}$), $I$ is the beam current (in A), and $a$ is the beam radius (in m). The stopping power has been obtained from the ESTAR database available from the NIST website. Supplementary Table 1 shows the values used to calculate the dose rate for 300 kV electrons in pure liquid water at the experimental currents used. The mean distance traveled has been calculated with the Chemsimul program using the diffusion coefficients[67] shown in Supplementary Table 2 and the average lifetime ($T_{1/2}$) calculated in Supplementary Table 3.

### Definition of a generalized GT equation
To model the kinetics of dissolution for arbitrarily shaped particles, we developed adaptation of the well-known Gibbs Thomson (GT) relation that is amenable to numeric evaluation. Unlike most treatments of GT that require the calculation of local surface curvatures, this

formulation is expressed in terms of generalized geometrical coordinates. For each coordinate, a driving force for perturbation is determined, thus enabling easy treatment of complex geometries without the need to explicitly determine surface curvatures. For simplicity, this generalization assumes that the surface tension and growth kinetics are independent of face orientation.

The GT relation is typically expressed in terms of the pressure drop across a curved interface. However, the underlying physics of the GT relation is based on an analysis of the change in particle surface area and volume that are associated with any perturbation of the interface (i.e., advance or retreat). This relationship can be generalized to consider arbitrary geometries by defining the particle shape in terms of a set of coordinates $\{q_i\}$. For each coordinate, we determine two derivatives: $\partial V/\partial q_i$ and $\partial S/\partial q_i$, which are the derivatives of particle volume, $V$, and surface area, $S$, associated with perturbations of each $q_i$. Using these derivatives, we can define a generalized pressure drop associated with each coordinate, where $\gamma$ is the surface tension:

$$\Delta p_i = \frac{\gamma \left(\frac{\partial S}{\partial q_i}\right)}{\left(\frac{\partial V}{\partial q_i}\right)} \quad (2)$$

Using standard approaches, the pressure drop can be cast in terms of a change in equilibrium concentration, by treating the solid phase as incompressible with a monomer unit volume of $v_m$, and by treating the aqueous solution as an ideal solution (which is valid if we limit ourselves to small perturbations about the bulk equilibrium concentration, $C_0$, such that the chemical potential can be expressed as $\mu = \mu_0 + \ln C/C_0$). Thus, we can relate each pressure drop to an equilibrium concentration as follows:

$$C_i = C_0 \exp \frac{v_m \Delta p_i}{k_b T} \quad (3)$$

The combination of the two equations above provides a generalization of the GT equation. When a particle is immersed in a solution with solute concentration $C_{sol}$, a particular coordinate is in equilibrium when $C_{sol} = C_i$. However, if the solution concentration, $C_{sol}$, differs from this value, then there will be a driving force to change $q_i$ by dissolution or growth.

In this study we seek to analyze the behavior of hematite rods and necks. Thus, we approximate the rod and neck geometry as a sequence of conical segments defined by the radii $r_i$, at regular spacing $\Delta x$ as shown in Supplementary Fig. 8. The volume ($V$) and surface areas ($S$) affected by each coordinate (comprising two conical segments) can be expressed by the following equations, from which the derivatives and associated pressures can be readily determined:

$$S_i = \pi(r_i + r_{i-1})\sqrt{(r_i - r_{i-1})^2 + \Delta x^2} + \pi(r_i + r_{i+1})\sqrt{(r_i - r_{i+1})^2 + \Delta x^2} \quad (4)$$

$$V_i = \pi(2r_i^2 + r_{i-1}^2 + r_{i+1}^2 + r_i r_{i-1} + r_i r_{i+1})/3 \quad (5)$$

## Kinetic model of dissolution

Having established the local thermodynamic driving force for growth and dissolution, we implement a kinetic model for particle shape evolution as follows: The rate of molecular attachment contributing to the growth of each coordinate is assumed to be $kC_{sol}S_i/2$, where $C_{sol}$ is the concentration of ions in solution, $S_i$ is the surface area of the volumes adjacent to each coordinate, $k$ is a rate constant with dimensions of attachment events per unit area, and the factor of ½ is

introduced since the influence of any given attachment event is shared between the two adjacent coordinates. The rate of molecular detachment is similarly given as $kC_iS_i/2$, where the solution concentration is replaced by the shape-dependent local equilibrium concentration. This gives a net rate of molecular addition of $k(C_{sol} - C_i)S_i/2$, such that growth associated with a given coordinate ceases when the local equilibrium concentration matches that of the solution. Then accounting for the volume of the attaching unit, the net rate of change for each coordinate is evaluated as follows:

$$\frac{\partial q_i}{\partial t} = \frac{1}{2} \frac{k v_m s_i (C_{sol} - C_i)}{\left(\frac{\partial V}{\partial q_i}\right)} \quad (6)$$

Thus, we obtain governing equations for the rate of change for each coordinate that can be readily calculated with SymPy symbolic math package in Python.

## Application of the model to experimental data: particles of diverse size and aggregates

The model also allows us to understand the different dissolution rates of rhombs and spindles. The dissolution rates for rhombic particles are correlated with the dissolution rate for similarly sized spheres. Faster dissolution seen for smaller rhombs and slower dissolution for larger rhombs. Importantly, the spindles dissolve much faster than comparatively sized rhombs, and whereas the rhombs are only slightly smaller at the end of the data series, the spindles are largely disintegrated. This dramatic difference indicates that spindle dissolution rate is linked to the size of the primary particles, rather than the total spindle size. With the parameters used here, here we show that spheres of 5 nm radius will dissolve in roughly the same timescale as the timescale for spindle dissolution.

## Application of the model to experimental data: dissolution of necks

Finally, we apply the model to investigate the dissolution of rod-shaped particles containing necks. Necks are typically unstable features, and thus there are two possibilities of evolution for a given neck: a neck may become shallower relative to the rod diameter (essentially self-healing), or the neck become deeper relative to the rod diameter (causing the neck to pinch off, such that the rod disintegrates into particles).

The impact of surface tension on neck evolution can be determined through the classical GT relation. Healing will be expected if the equilibrium concentration associated with the neck is lower than for the surrounding rod while deepening will be expected if the equilibrium concentration associated with the neck is higher than for the surrounding particle. The classical GT theory relates the local solubility to radius of curvature. In the deepest incision of a neck, we encounter a saddle geometry with two radii of curvature: one positive and one negative: whether a neck will deepen or heal depends on these radii.

Our simulations treat two representative types of necks: sharp and narrow incisions, vs broad undulations. Each neck is represented as a sinusoidal perturbation in a rod-shaped particle. The simulations show that after a brief relaxation, the broad undulations become unstable and deepen to pinch off. In contrast, the high negative curvature in the sharp incisions actually drives a dramatic healing process during which the neck becomes shallower and broader. Only later does the neck become broad enough that it begins to deepen and pinch off. Thus, we anticipate that surface-tension driven necking will be identifiable by the gradually deepening of broad undulations.

Although some necking events due occur to form via broad undulations as predicted, many of the necking events appear to occur via sharp incisions that become even sharper in the last moments as

the neck pinches off, a behavior that is not consistent with the predictions of the surface-tension driven model. This suggests that a more complex mechanism is driving necking in these cases.

The occurrence of necking via sharp incisions can be most readily understood as being driven by preexisting, high-energy defects such as grain boundaries (GB). We can model this effect in the generalized GT simulations by treating the grain boundary as a surface with an associated surface energy. Typical grain boundary energies (GBEs) are on order of 0.26 to 1.47 J m$^{-2}$. By associated a GB with one of the radial coordinates, with a surface area of $S_{GB} = \pi r_{GB}^2$, we can simulate the influence of the GB on necking behavior. These simulations show that a GB can drive necking via a sharp incision, rather than a broad undulation. The angle of the deepening incision is related to the balance of the GBE and particle surface energy, and the incisions become sharper the as the GBE becomes larger relative to the surface energy. Especially rapid incision occurs when the GBE > 2$\gamma$, in which case the particle is thermodynamically favored to completely separate at the GB.

## Reporting summary

Further information on research design is available in the Nature Portfolio Reporting Summary linked to this article.

## Data availability

The data supporting the findings of this study are available from the corresponding author upon request.

## Code availability

The codes used for the findings of this study are available from the corresponding author upon request.

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

## Acknowledgements

This material is based upon work supported by the US Department of Energy (DOE), Office of Science, Office of Basic Energy Sciences, Chemical Sciences, Geosciences, and Biosciences Division, Geosciences Program at Pacific Northwest National Laboratory (PNNL) through FWP 56674. High resolution TEM and STEM imaging and DFT simulations were performed in the Environmental and Molecular Sciences Laboratory, a DOE Office of Science User Facility at PNNL sponsored by the Office of Biological and Environmental Research. PNNL is a multi-program national laboratory operated for DOE by Battelle under Contract No. DE-AC05–76RL01830.

## Author contributions

J.J.D.Y. supervised the research. G.Z. synthesized the materials and performed the TEM studies and analyses. B.A.L. carried out the analytic modeling. M.S. performed the radical species calculation. X.L. helped with some TEM data analyses. G.Z. and M.Z. performed the ICP measurements. K.M.R. engaged in discussions extensively. All authors contribute to the writing.

## Competing interests

The authors declare no competing interests.
