## [Peer Review File · Nature Communications]

Crystal Dissolution by Particle DetachmentREVIEWER COMMENTS

Reviewer #1 (Remarks to the Author):

This is a very interesting and timely study asking the question if particle-based attachment has a counterpart in dissolution in analogy to classical crystal growth / dissolution by attachment / detachment of atomic/molecular building units. The applied in-situ TEM techniques are suitable to investigate this interesting question, because it relies on the time dependent observation of particles in solution. Comparing the dissolution of hematite single crystals and hematite mesocrystals, the authors found that the mesocrystals dissolve much faster and by detachment of particles. This is also beautifully demonstrated in a dissolution experiment on a mixture of both particle types with similar size. The study is technically performed on a high level and the obtained results are well justified. They are also important given the assumption that materials assembled by particle attachment are much more frequent than previously believed so that their dissolution behavior also needs to be understood. Therefore, I support publication of this work.

However, I would recommend that the authors add a little bit of discussion on the fact that the transition between a mesocrystal and a single crystal is more or less continuous depending on the degree of crystallographic fusion of the nanoparticles in the mesocrystal (Ref. 17 in the paper for example or doi.org/10.1021/jp068813i for a time resolved neutron scattering study). In other words, a mesocrystal can ripen to a single crystal but if it is still a mesocrystal, it can easily dissolve into a nanoparticle dispersion and be recrystallized as for example shown in doi.org/10.1002/chem.202002873 for magnetite mesocrystals (in that case stabilized by organic molecules to slow down / prevent crystallographic fusion)

As the authors say in the last sentence of their paper, this study is a starting point for the consideration of dissolution by particle detachment and therefore, this extended discussion will be helpful for the reader. The dissolution of a single crystal is well documented and described – also in this study. What is unclear though is to what extent, the individual nanoparticles in the original hematite mesocrystal have already fused to a single crystal. Looking at the beautiful videos, here are particles easily detaching in the beginning leaving a branched skeleton, which appears to be single crystalline. This might be indicated in Fig. S1

where at least for center and middle of the particle its mass stays rather constant until frame 5, then drops quickly until frame 9, possibly by detachment of entire particles. Afterwards, it drops by a lower rate and here, the particles appear to be well connected so that their detachment is slower.

For these single crystalline particle skeletons, the authors observe the formation of necks and observe their development until a particle detaches. This can be described by their thermodynamic / kinetic model, which I propose to move into the paper.

What remains open in the discussion is how the degree of particle fusion in the mesocrystal can be taken into account and this is certainly of importance to finally get to a good description and understanding of what is going on. Perhaps, a first step on this way could be to take a look at the dissolution of hematite mesocrystals after different annealing times like one directly after synthesis for which fast dissolution would be expected and another sample, which was stored for a certain time. That could give a qualitative insight. It would be ideal if the amount of individual vs. fused nanoparticles could be determined, because this would then allow to correlate the dissolution results also to this number. However, I do not have a suggestion how to obtain this.

Alternatively, a system would be ideal, where the particle attachment as well as detachment could be observed in situ with the advanced TEM techniques of the authors. That would even allow for cycling of conditions to determine both – rate of particle attachment vs. rate of detachment in mesocrystals. A system, which might be well suitable for such an investigation is Prussian Blue Analogue mesocrystals like nickelhexacyanoferrate. They are available as monodisperse cubes in sizes from 20 – 150 nm and they self-assemble to mesocrystals within a few h due to electrostatic shielding by the ionic strength of the surrounding solution (doi.org/10.1021/acs.jpcc.1c08284). Particle detachment / attachment should be tunable by the ionic strength, the particles do not dissolve in water but in alkaline solution (doi.org/10.3390/nano11102756). Whether or not the nanoparticles in the mesocrystal can fuse crystallographically was not yet reported but appears feasible.

Reviewers #2 and #3, who reviewed jointly (Remarks to the Author):

The authors present experimental results for dissolution by particle detachment (DPD), which is the inverse of crystallization by particle attachment (CPA). The CPA theory was published by the same group in 2015 contrasting the classical monomer-by-monomer crystallization theory.

The SI videos showing DPD are impressive and provide new in-situ evidence about dissolution processes for branched porous nanoparticles. The data treatment retrieving dissolution rates and a focus on the necking process with subsequent particle detachment are interesting. And while some plots are poorly described and need enhancement (see below), such deficiencies could certainly be fixed in a revision. Nevertheless, the main finding, that nanoparticles that grow via CPA likely dissolve following DPD, is rather expected in light of prior discussion of solute dissolution in the literature (see next paragraph). No further scientific insights or predictions arising from the observation are reported or discussed. For these reasons, while certainly of interest to the crystallization and materials science community, the manuscript does not fulfill the requirements of Nature Communications in terms of novelty and broad impact. I therefore recommend publication in a more specialized journal.

Aggregation/agglomeration theories opposing monomer-by-monomer crystallization exist since Smoluchowski [1]. This pioneering work gave rise to the field of population balance models (PBM) [2]. Aggregation/agglomeration (or CPA) tends to generate fractal-type crystals. Including limitation of particles to be aggregated to the main crystal because of diffusion, which is the basic process of forming fractals, leads to the theory of diffusion limited aggregation (DLA) [3]. Dissolution has already been included in DLA-frameworks [4]. Furthermore, there exists evidence for the detachment of clusters during local dissolution [5]. Finally, there appears to be a relationship to the "chunk effect" discussed in this paper [6], even though research in this area goes back to the 1960s. As these examples demonstrate, there is much prior research, all of which is ignored. In light of these and related works, I would expect deeper insight than a mere observation of the DPD phenomenon that expands our scientific understand further.

[1] M. von Smoluchowski, Versuch einer mathematischen Theorie der Koagulationskinetik kolloider Lösungen, Z. für Phys. Chemie, 92, 129, (1917)

[2] D. Ramkrishna, Population Balances: Theory and Applications to Particulate Systems in Engineering, 1st ed., San Diego (2000).

[3] T.A. Witten, L.M. Sander, Diffusion-Limited Aggregation, a Kinetic Critical Phenomenon, Phys. Rev. Lett. 47, 1400 (1981)

[4] Z. Xu, P. Meakin, Phase-field modeling of two-dimensional solute precipitation/dissolution: Solid fingers and diffusion-limited precipitation, J. Chem. Phys. 134, 044137 (2011)

[5] R.G. Buchheit et al Local Dissolution Phenomena Associated with S Phase (Al₂CuMg) Particles in Aluminum Alloy 2024-T3 J. Electrochem. Soc. 144, 2621 (1999)

[6] F.A. Silveira, F.D.A. Aarão Reis, Detachment of non-dissolved clusters and surface roughening in solid dissolution, Electrochim. Acta 111, 1 (2013)

Further comments and questions:

1) The manuscript jumps immediately to the results after the abstract/introduction paragraph. In light of existing prior work, only some of which is discussed above, an extensive introduction presenting the state of the art in chemical dissolution is a necessity.

2) All theory has been moved to SI and only the experimental evidence remains in the main text. This is unfortunate, as a theoretical discussion would give the manuscript more substance. As long as the theory is not in the main text, the manuscript is merely a discussion of the experimental findings. I recommend moving some of the theory into the main text.

3) Lines 57-59: "While the body of each particle dissolves at a more-or-less constant rate

(Fig. 2c, open symbols), the necks shrink at an accelerating rate with time". This acceleration makes sense if one analyses the neck diameter change (L), as dissolution rate is proportional to surface area (L^2). But would that make sense though if one analyses the volume (L^3) of the dissolved crystal? Note that dissolution is a surface-to-volume problem, i.e., the volume of the crystal is dissolved through its surface.

4) Lines 59-60: "upon shrinking to a diameter of ~ 1.0 nm, the necks disintegrate". Could one say this is evidence that the neck cannot sustain anymore the kinetic energy of the system, thus breaking the neck? Notice that the critical value reported for neck fraction is close to 1 nm. How many atomic bonds fit in a distance of 1 nm?

5) Figure 2c: What do the colors represent?

6) Lines 92-93: "we then estimate that at pH 7 the mineral surface accessible to the diffusion ranges of eaq^- and O_2^- are within 130 nm and 200 nm, respectively". What does "mineral surface" mean, and what do these distances represent?

7) Figure 3i-k: I suppose the different lines in the same plots represent different times. Please make the advancement of time clear in the plots.

8) Figure 4: "Quantitative analyses of the dissolution and modeling". Do you mean "of the dissolution model"? In (b), are the red/orange and blue colors swapped? The blue particle appears to be the smallest in (e), while red/orange is the largest. What is the meaning of the colors and squares/circles in (c)? The y-axis title of (c) should be improved. "Fe amount" does not say much.

9) Minor: In caption of Figure 1, use capital letter in "difference".

10) Gibbs(-)Thom(p)son, with or without hyphen, with or without "p"? Various spelling variants are used throughout the text.

Reviewer #4 (Remarks to the Author):

This manuscript proposed a process of dissolution by particle detachment (DPD), which is an inverse counterpart of crystallization by particle attachment (CPA). Through the in-situ liquid phase TEM technology, the authors directly observed the DPD phenomenon of typical mesocrystals of hematite Fe_2O_3 with spindle-shaped morphology. In short, this work provides promising research directions for DPD applications for various fields, which can be published in Nature Communications by addressing the following issues.

1. Although the authors verified on the material they used that DPD is the inverse process of CPA, the universality of the DPD phenomenon on the materials formed by CPA still needs to be further explored. The correlation between the DPD behavior and the structure of small crystals in mesoscopic crystals is an example, while the authors also indicated that the dissolution rates of neck and body in the sharp-neck structures are very close (Fig. 4l). Therefore, some relevant expressions need to be reconsidered.
2. Why the dissolution rates of rhHm and spHm measured by ICP-OES are fast at the dissolution beginning? More explanations are needed.
3. Observing a nanostructural transformation process may not be a research goal, but what needs to be known is how to connect the DPD phenomenon with its potential applications. Related discussion and outlook should be included to this work.
4. Whether the DPD phenomenon is universal, if it is only a very unique system, then this phenomenon may lose its great significance of research. Can we recklessly speculate that this phenomenon may not exist in a system with excellent crystallinity? Besides the unique structure of Fe_2O_3 , is it possible to observe it in other systems?

Reviewer #5 (Remarks to the Author):

This study proposes that the inverse of crystallization by particle attachment (CPA, also commonly referred to as “non-classical crystallization”) is dissolution by particle detachment (DPD). The authors provide a demonstration of this phenomenon by examining and comparing the dissolution behavior of hematite nanocrystals with two distinct morphologies (rhombohedral vs. spindle-shaped mesocrystals). They demonstrate that the mesocrystal morphology (a consequence of growth via CPA) exhibits significantly higher rates of dissolution compared with the compact rhombohedral nanocrystals. They suggest that

crystals that form via CPA processes may dissolve via DPD due to distinct characteristics of their nanostructure. Specifically, in this case they attribute the enhanced dissolution to the localized impacts of lattice strain and particle curvature, both of which were consequences of the (near) oriented attachment of nanosized precursors that produced branch-like structures.

The work is of excellent quality, is significant in advancing understanding of mineral dissolution, and is novel. They effectively use advanced in situ (and ex situ) imaging methods combined with other techniques to link nanoscale characteristics to macroscopic properties.

I don't see any need for major revisions and support publishing with only minor changes. One aspect that I think should be emphasized (or made a bit clearer) relates to the importance of dissolution by monomer detachment that is occurring in systems involving DPD. In other words, it is generally accepted that classical and non-classical crystallization (i.e., CPA) are not mutually exclusive, even if one dominates the other for a particular material or set of conditions. I think that the authors are demonstrating that this is also the case in the present example (so-called "necking") during DPD, but would like the text to be a bit more explicit.

In regards to the specific review questions:

What are the noteworthy results?

The concept of DPD is novel and has not been described in detail in the literature. The results provided in the manuscript are effective at demonstrating DPD.

Will the work be of significance to the field and related fields? How does it compare to the established literature? If the work is not original, please provide relevant references.

The work will be of significance to many different fields. There have been many decades of research on the dissolution behavior of minerals in the context of mineral science, materials science, etc. but many important questions remain, in particular in cases where the dissolution can not be readily or entirely explained using a classical theory involving monomer-by-monomer detachment.

Does the work support the conclusions and claims, or is additional evidence needed?

Yes

Are there any flaws in the data analysis, interpretation and conclusions? Do these prohibit publication or require revision?

No

Is the methodology sound? Does the work meet the expected standards in your field?

Yes

Is there enough detail provided in the methods for the work to be reproduced?

Yes

Reviewer1:

This is a very interesting and timely study asking the question if particle-based attachment has a counterpart in dissolution in analogy to classical crystal growth / dissolution by attachment / detachment of atomar/molecular building units. The applied in-situ TEM techniques are suitable to investigate this interesting question, because it relies on the time dependent observation of particles in solution. Comparing the dissolution of hematite single crystals and hematite mesocrystals, the authors found that the mesocrystals dissolve much faster and by detachment of particles. This is also beautifully demonstrated in a dissolution experiment on a mixture of both particle types with similar size. The study is technically performed on a high level and the obtained results are well justified. They are also important given the assumption that materials assembled by particle attachment are much more frequent than previously believed so that their dissolution behavior also needs to be understood. Therefore, I support publication of this work.

Reply: We would like to thank the reviewer for acknowledging the significance of our study by noting “very interesting and timely study” of our work and “the applied in-situ TEM techniques are suitable to investigate this interesting question”. We also appreciate the reviewer for recognizing that “the study is technically performed on a high level and the obtained results are well justified”, “they are also important given the assumption that materials assembled by particle attachment are much more frequent than previously believed so that their dissolution behavior also needs to be understood”, and “support publication of this work”.

However, I would recommend that the authors add a little bit of discussion on the fact that the transition between a mesocrystal and a single crystal is more or less continuous depending on the degree of crystallographic fusion of the nanoparticles in the mesocrystal (Ref. 17 in the paper for example or doi.org/10.1021/jp068813i for a time resolved neutron scattering study). In other words, a mesocrystal can ripen to a single crystal but if it is still a mesocrystal, it can easily dissolve into a nanoparticle dispersion and be recrystallized as for example shown in doi.org/10.1002/chem.202002873 for magnetite mesocrystals (in that case stabilized by organic molecules to slow down / prevent crystallographic fusion)

Reply: We appreciate the suggestions and completely agree with the comment. Indeed, in other work we have investigated the effect of synthesis temperature and time on the coarsening of hematite spindles towards more compact rhombohedral crystals (Zhu and De Yoreo. J. Electrochem. Soc. 169, 102510, 2022). There is a continuum from one to the other and where a given product lies on that continuum depends on these two factors. To address this comment, we have added the following to the introduction and discussion of the main text:

On page 2, in an expanded introduction “Although research on CPA has drawn significant interest in recent years²¹, whether, by analogy, the reverse process of dissolution by particle detachment (DPD) occurs and what determines its contribution to the overall dissolution rate relative to concomitant classical monomer-by-monomer dissolution has received little attention. One rare example is the “chunk effect” that has been observed in electrochemical corrosion^{3,35-38}, wherein the disintegration of polycrystalline materials into individual particles is believed to explain anomalously high dissolution rates. However, the polycrystals do not arise from CPA and an electric field is usually involved to generate the effect. None the less, past observations that OA-generated mesocrystals in which the primary particles are separated by intervening hydration layers can disintegrate into the primary particles during sublimation of the hydration layers²³ provides evidence that mesocrystalline solids grown via CPA can undergo DPD. However, whether or not a similar process is possible when the constituent particles have fused to form a continuous crystal lattice

has not been explored. Understanding the mechanisms of DPD will be important to modeling the durability and reactivity of diverse materials, especially for the complex hierarchically structured materials that are frequently synthesized using CPA.”

In a new paragraph near the end of the discussion section, to address this comment and comments of other reviewers, we have added, “The extent to which rates of DPD and resultant dissolution morphologies differ from those seen in compact crystals grown by the classical process of ions or ion clusters addition to steps on faceted crystal surfaces depends on inherent materials parameters, such as solubility, surface tension, and elastic constants (via the grain boundary energy), as well as external factors of temperature and incubation time. While the expressions used in the model presented here are general and include the inherent materials parameters explicitly, the external factors are only implicitly incorporated into the analysis through the as-observed morphology. Higher temperatures and/or longer times spent in solution near the saturation point lead to more extensive coarsening, which increases average particle size, reduces curvature, relieves misalignment and corresponding strain at the boundaries, and eliminates concave regions^{27,58,59} to an extent that also depends on the atomic mobility of the material. The implication is that the extent to which DPD dominates the dissolution process for a given material system cannot be predicted simply based on materials parameters alone; rather, either priori knowledge of the CPA dynamics that led to formation of the mesocrystals or observational data on the resulting morphology and distribution of grain boundaries is required. Conversely, the influence of both internal and external factors implies an ability to tune the speed of the dissolution using methods to control the primary particle size and strain, which could be applied in areas like corrosion engineering, nutrient release, and drug delivery. Future work should explore the factors that define the grain-boundary energies between particles in mesocrystals and how this influences the degree to which DPD occurs.”

As the authors say in the last sentence of their paper, this study is a starting point for the consideration of dissolution by particle detachment and therefore, this extended discussion will be helpful for the reader. The dissolution of a single crystal is well documented and described – also in this study. What is unclear though is to what extent, the individual nanoparticles in the original hematite mesocrystal have already fused to a single crystal. Looking at the beautiful videos, here are particles easily detaching in the beginning leaving a branched skeleton, which appears to be single crystalline. This might be indicated in Fig. S1 where at least for center and middle of the particle its mass stays rather constant until frame 5, then drops quickly until frame 9, possibly by detachment of entire particles. Afterwards, it drops by a lower rate and here, the particles appear to be well connected so that their detachment is slower.

Reply: We appreciate the reviewer’s observation that the skeleton structure revealed during dissolution is more crystalline, and that understanding this transformation into a more crystalline structure is an important area of research. The skeleton and its crystallinity are not easily discernible as they are embedded in the mesocrystal, which is a feature we did not explore in our previous work (Zhu, Guomin, et al. "Self-similar mesocrystals form via interface-driven nucleation and assembly." *Nature* 590.7846 (2021): 416-422.). As shown in Fig. 3e-g, the higher crystallinity in the skeleton suggests that there is crystallographic fusion during synthesis. Our recent work demonstrates that this crystallographic fusion is dependent on the temperature and time during and after the synthesis (Zhu, Guomin, and James J. De Yoreo. "Hematite Crystallization from a Two-Line Ferrihydrite Suspension." *Journal of The Electrochemical Society* 169.10 (2022): 102510.). We showed that after aging for an extended period, the spindle structure begins to develop this more crystalline skeleton and the characteristic size of primary particles starts to increase, providing evidence that Ostwald ripening or some other mechanism may be responsible for the transition from

mesocrystal to a more single crystalline state (see figure below). At sufficiently high temperatures, we have even observed complete transformation of the mesocrystal to a single crystal with a rhombohedral shape.

Figure. Effect of aging time on the hematite mesocrystal structure. Aging of the hematite solution for 10 hours vs. 270 hours. The size of the primary particles grows and the spindle starts to develop the skeleton structure that is more resistant to dissolution than the exterior regions that form at later times.

Here are the changes made to address this point:

On page 7, “As mentioned previously, the continuous nature of the crystal lattice is apparent; this misorientation between grains appears to be facilitated by dislocations, as for a low-angle grain boundary (Fig. 3f). The grain boundary — i.e., strain — energy at these contacts can be estimated from the Shockley-Read equation, which dictates that it increases linearly with respect to the degree of grain mismatch below a certain angle²². The greater recalcitrance to dissolution by the interior skeleton suggests that this region has coarsened to reduce the defect density, coarsen the particles, and eliminate concavities in the branches.”

On page 9, “The extent to which rates of DPD and resultant dissolution morphologies differ from those seen in compact crystals grown by the classical process of ions or ion clusters addition to steps on faceted crystal surfaces depends on inherent materials parameters, such as solubility, surface tension, and elastic constants (via the grain boundary energy), as well as external factors of temperature and incubation time. While the expressions used in the model presented here are general and include the inherent materials parameters explicitly, the external factors are only implicitly incorporated into the analysis through the as-observed morphology. Higher temperatures and/or longer times spent in solution near the saturation point lead to more extensive coarsening, which increases average particle size, reduces curvature, relieves misalignment and corresponding strain at the boundaries, and eliminates concave regions^{27,58,59} to an extent that also depends on the atomic mobility of the material. The implication is that the extent to which DPD dominates the dissolution process for a given material system cannot be predicted simply based on materials parameters alone; rather, either priori knowledge of the CPA dynamics that led to formation of the mesocrystals or observational data on the resulting morphology and distribution of grain boundaries is

required. Conversely, the influence of both internal and external factors implies an ability to tune the speed of the dissolution using methods to control the primary particle size and strain, which could be applied in areas like corrosion engineering, nutrient release, and drug delivery. Future work should explore the factors that define the grain-boundary energies between particles in mesocrystals and how this influences the degree to which DPD occurs.”

On page 9, “Here we add to the list differences, the dissolution behavior of crystals generated through CPA from that of monolithic crystals, with consequences for a wide range of phenomena, such as mineral weathering, reactive transport, recrystallization, release of nutrients, and drug delivery. The analyses presented here provide a starting point for predicting how DPD influences such phenomena. The exact role of DPD needs to be carefully studied to understand how it both depends on and impacts microstructural evolution and to connect its characteristics to the above phenomena.”

For these single crystalline particle skeletons, the authors observe the formation of necks and observe their development until a particle detaches. This can be described by their thermodynamic / kinetic model, which I propose to move into the paper.

Reply: We have moved much of our modeling work into the main text, as suggested.

On page 7 we expanded the text to read, “To understand the roles of particle misorientation and interfacial curvature, we developed a general model for dissolution that accounts for the impact of surface energy, γ_s , grain boundary energy, γ_{GB} , and particle shape. This model (explained in detail in the Supplemental Materials) considers how interfaces (including the particle surface area, S_s , and grain boundary area, S_{GB}) produce a Laplace pressure that changes the thermodynamic driving force for dissolution. Our model calculates the Laplace pressure in terms of generalized coordinates, and thus allows us to easily treat the dissolution of arbitrary shapes. Here, we model necked particles as a series of conical segments, defined by the radii, r_i , where each segment meets. In brief summary, the Laplace pressure associated with a coordinate, q_i , is obtained from the ratio of the derivative in interfacial energy with respect to that coordinate ($\gamma \partial S / \partial q_i$) to the derivative in volume ($\partial V / \partial q_i$). The Gibbs-Thomson equation then allows us to convert the Laplace pressure into changes in local solubility about each segment. Increased solubilities are modeled as an enhancement in the local ion detachment rate (per unit area), which provides the basis for a kinetic model that predicts the evolution of particle shape.

The solubilities predicted by our model agree with standard implementations of the Gibbs-Thomson equation, which calculate the Laplace pressure in terms of local Gaussian curvature. Simple spherical and cylindrical particles have a positive surface curvature that leads to enhanced solubility for smaller particles. Thus, smaller spherical particles dissolve at an enhanced rate (Fig. 4a,b). However, the necks between particles display a combination of positive curvature (which favors deeper incision at the neck) and negative curvature (which favors neck healing). Competition between these effects determines the evolution of neck shape.

Our models predict that the negative curvature dominates for sharp necks, which typically cause the neck to heal. Thus, most initially-sharp necks will quickly disappear and not contribute significantly to particle detachment. (Fig. 3i,l). For wider necks, the positive curvature becomes dominant. These necks exhibit some initial healing, but ultimately trend towards more rapid dissolution at the neck than at the adjacent branch, leading to detachment (Fig. 3j,l). However, the neck profile always remains wide and gently curving, and by the time a neck finally separates, the rest of the branch has significantly dissolved too. Thus, this pathway is also unlikely to produce long-lived individual particles. A third scenario is given by a branch

with no initial neck but with a high energy grain boundary (Fig. 3k). The grain boundary also contributes to the Laplace pressure, and causes a sharp incision to form that leads to rapid neck dissolution and break-up (Fig. 3k,l).

Thus, these three scenarios exhibit distinct dissolution dynamics (Fig. 3l). More complex dynamics could be expected in a real neck configuration where curvature is variable and is accompanied by varying levels of grain boundary energy. These three scenarios and their combinations provide a rationale for why only certain regions of neck break-up are observed *in situ*. The TEM results (Fig. 2a and b) could be explained by a branch that contains no initial neck in the beginning (0.0s), but where a high-energy grain boundary causes the necking to initiate, and gradually form a wider and deeper neck (Fig. 2a, 4.8s and Fig. 2b 9.6s, indicated by the arrow). Eventually, the rapid dissolution at the neck leads to the break-up and cause the particles to detach.

What remains open in the discussion is how the degree of particle fusion in the mesocrystal can be taken into account and this is certainly of importance to finally get to a good description and understanding of what is going on. Perhaps, a first step on this way could be to take a look at the dissolution of hematite mesocrystals after different annealing times like one directly after synthesis for which fast dissolution would be expected and another sample, which was stored for a certain time. That could give a qualitative insight. It would be ideal if the amount of individual vs. fused nanoparticles could be determined, because this would then allow to correlate the dissolution results also to this number. However, I do not have a suggestion how to obtain this.

Reply: We agree with the reviewer that the degree of particle fusion in the mesocrystal would certainly play a big role in the DPD process. A system in which each state of the transformation is stable and could be easily characterized would allow us to better understand the degree of particle fusion on the phenomena. However, the main challenge is that the spindles take time to develop and coarsening is occurring during that time so that the inner regions — i.e., the skeleton — have fewer defects and less curvature in the regions where the particles original attached to one another. We should point out, however, that the model itself accounts for all of this, because it is based on the curvature, size, and strain, regardless of whether they have been diminished during coarsening. If one could create a 3D tomographic map of spindle morphology and strain distribution, the inhomogeneity in degree of crystalline perfection due to the temporal evolution could be completely taken into account. We thank the reviewer for bringing up this interesting direction for the future work.

Alternatively, a system would be ideal, where the particle attachment as well as detachment could be observed in situ with the advanced TEM techniques of the authors. That would even allow for cycling of conditions to determine both – rate of particle attachment vs. rate of detachment in mesocrystals. A system, which might be well suitable for such an investigation is Prussian Blue Analogue mesocrystals like nickelhexacyanoferrat. They are available as monodisperse cubes in sizes from 20 – 150 nm and they self-assemble to mesocrystals within a few h due to electrostatic shielding by the ionic strength of the surrounding solution (doi.org/10.1021/acs.jpcc.1c08284). Particle detachment / attachment should be tunable by the ionic strength, the particles do not dissolve in water but in alkaline solution (doi.org/10.3390/nano11102756). Whether or not the nanoparticles in the mesocrystal can fuse crystallographically was not yet reported but appears feasible.

Reply: We thank the reviewer for this recommendation. The nickel hexacyanoferrate mesocrystal system sounds fascinating system and appears to exhibit mesocrystal evolution at room temperature at a speed that is reasonable for the proposed study.

Reviewers #2 and #3, who reviewed jointly (Remarks to the Author):

The authors present experimental results for dissolution by particle detachment (DPD), which is the inverse of crystallization by particle attachment (CPA). The CPA theory was published by the same group in 2015 contrasting the classical monomer-by-monomer crystallization theory.

The SI videos showing DPD are impressive and provide new in-situ evidence about dissolution processes for branched porous nanoparticles. The data treatment retrieving dissolution rates and a focus on the necking process with subsequent particle detachment are interesting. And while some plots are poorly described and need enhancement (see below), such deficiencies could certainly be fixed in a revision. Nevertheless, the main finding, that nanoparticles that grow via CPA likely dissolve following DPD, is rather expected in light of prior discussion of solute dissolution in the literature (see next paragraph). No further scientific insights or predictions arising from the observation are reported or discussed. For these reasons, while certainly of interest to the crystallization and materials science community, the manuscript does not fulfill the requirements of Nature Communications in terms of novelty and broad impact. I therefore recommend publication in a more specialized journal.

Reply: We thank the reviewer for noting that the “*The SI videos showing DPD are impressive and provide new in-situ evidence*” and “*The data treatment retrieving dissolution rates and a focus on the necking process with subsequent particle detachment are interesting*”.

We also appreciate the insights and suggestions from the reviewers in this comment and later ones. However, we respectfully disagree with the characterization of the work as lacking novelty and broad impact. With regard to novelty, there are two aspects to consider. One is the concept of DPD itself. We will hold off on addressing this aspect in the current response, because it is largely addressed in our response to the next paragraph of the review, which follows immediately after this response. We will simply say at this point, in anticipation of the next response, that what we report here is extremely different from the three examples given in the next paragraph of the review — disintegration of DLA fractal aggregates, the chunk effect, and dissolution through detachment of clusters — none of which anticipate DPD of mesocrystalline solids grown through CPA. Again, the details are given below.

The second aspect of novelty is the elements of the study itself. The reviewers have already noted the novelty of the data and its treatment. Our state-of-the-art in situ experiments were carefully designed and executed to probe the dissolution kinetics and pathway. This is the first time that necking and particle detachment processes have been captured in situ and here we do so at the nanometer scale. However, we also present a novel theoretical model based on the Gibbs-Thomson relation that incorporates curvature and strain into expressions for dissolution of necked branches for a range of scenarios. Thus, we feel that both the elements of the study and, as shown below, the general concept are highly novel.

With respect to the issue of broad impact, we are surprised at the comment, given the realization over the past decade of how widespread the growth of crystals through particle attachment processes are, particularly oriented attachment in which the attaching particles are crystallographically coaligned or nearly so.

Oriented attachment has now been widely observed in semiconductors,¹⁻⁴ metals,^{5, 6} silicates,^{7, 8} oxides,⁹⁻¹⁹ fluorides,²⁰ carbonates,¹² organic compounds,²¹ peptides,²² and proteins.²³ This is the class of crystals addressed in our manuscript. It seems to us that a manuscript demonstrating that the dissolution process is dramatically altered by the unique architecture arising from this process and that the dissolution rate can be nearly an order of magnitude faster as a consequence, while also providing a physical model for predicting dissolution based on a knowledge of morphology and interfacial strain for such a broad range of materials seems quite impactful and of great interest to the material science, geoscience, materials chemistry, electrochemistry, and corrosion science communities alike.

1. M. P. Boneschanscher, W. H. Evers, J. J. Geuchies, T. Altantzis, B. Goris, F. T. Rabouw, S. A. P. van Rossum, H. S. J. van der Zant, L. D. A. Siebbeles, G. Van Tendeloo, I. Swart, J. Hilhorst, A. V. Petukhov, S. Bals and D. Vanmaekelbergh, *Science*, 2014, **344**, 1377-1380.
2. K. S. Cho, D. V. Talapin, W. Gaschler and C. B. Murray, *J. Am. Chem. Soc.*, 2005, **127**, 7140-7147.
3. M. Kuno, *Phys. Chem. Chem. Phys.*, 2008, **10**, 620-639.
4. Z. G. Li, J. H. Sui, X. L. Li and W. Cai, *Langmuir*, 2011, **27**, 2258-2264.
5. J. M. Yuk, J. Park, P. Ercius, K. Kim, D. J. Hellebusch, M. F. Crommie, J. Y. Lee, A. Zettl and A. P. Alivisatos, *Science*, 2012, **336**, 61-64.
6. H. G. Liao and H. M. Zheng, *J. Am. Chem. Soc.*, 2013, **135**, 5038-5043.
7. T. M. Davis, T. O. Drews, H. Ramanan, C. He, J. S. Dong, H. Schnablegger, M. A. Katsoulakis, E. Kokkoli, A. V. McCormick, R. L. Penn and M. Tsapatsis, *Nat. Mater.*, 2006, **5**, 400-408.
8. A. I. Lupulescu and J. D. Rimer, *Science*, 2014, **344**, 729-732.
9. H. Cölfen, in *Handbook of biomineralization: Biological aspects and structure formation*, ed. E. Bauerlein, Weinheim, Germany, 2008, pp. 39-64.
10. J. Baumgartner, A. Dey, P. H. H. Bomans, C. Le Coadou, P. Fratzl, N. Sommerdijk and D. Faivre, *Nat. Mater.*, 2013, **12**, 310-314.
11. N. D. Burrows, C. R. H. Hale and R. L. Penn, *Cryst. Growth Des.*, 2013, **13**, 3396-3403.
12. M. H. Nielsen, D. S. Li, H. Z. Zhang, S. Aloni, T. Y. J. Han, C. Frandsen, J. Seto, J. F. Banfield, H. Cölfen and J. J. De Yoreo, *Microsc. Microanal.*, 2014, **20**, 425-436.
13. R. L. Penn, *J. Phys. Chem. B*, 2004, **108**, 12707-12712.
14. R. L. Penn and J. F. Banfield, *Science*, 1998, **281**, 969-971.
15. M. Raju, A. C. T. van Duin and K. A. Fichthorn, *Nano Lett.*, 2014, **14**, 1836-1842.
16. V. M. Yuwono, N. D. Burrows, J. A. Soltis and R. L. Penn, *J. Am. Chem. Soc.*, 2010, **132**, 2163-2165.
17. H. Z. Zhang, J. J. De Yoreo and J. F. Banfield, *ACS Nano*, 2014, **8**, 6526-6530.
18. D. Li, F. Soberanis, J. Fu, W. Hou, J. Wu and D. Kisailus, *Crystal Growth & Des.*, 2013, **13**, 422-428.
19. G. Zhu, M. L. Sushko, J. S. Loring, B. A. Legg, M. Song, J. A. Soltis, X. Huang, K. M. Rosso and J. J. De Yoreo, *Nature*, 2021, **590**, 416-422.
20. A. B. Bard, X. Z. Zhou, X. J. Xia, G. M. Zhu, M. B. Lim, S. M. Kim, M. C. Johnson, J. M. Kollman, M. A. Marcus, S. R. Spurgeon, D. E. Perea, A. Devaraj, J. Chun, J. J. De Yoreo and P. J. Pauzauskie, *Chem. Mater.*, 2020, **32**, 2753-2763.
21. S. L. Burkett and M. E. Davis, *J. Phys. Chem.*, 1994, **98**, 4647-4653.
22. J. J. Chen, E. B. Zhu, J. Liu, S. Zhang, Z. Y. Lin, X. F. Duan, H. Heinz, Y. Huang and J. J. De Yoreo, *Science*, 2018, **362**, 1135-1139.
23. A. E. S. Van Driessche, N. Van Gerven, P. H. H. Bomans, R. R. M. Joosten, H. Friedrich, D. Gil-Carton, N. A. J. M. Sommerdijk and M. Sleutel, *Nature*, 2018, **556**, 89-94.

To address this comment, we have added text to the manuscript related to the novelty our study and modeling, and added clear messages regarding the new scientific insights, predictions and breadth of impact.

Page 1-2: “Laboratory studies of dissolution kinetics in many fields^{2-5,16-18}, which are typically performed by monitoring rates of material release into solution or loss from material surfaces, have long pointed to the importance of quantifying the evolution of reactive surface area. Doing so provides a basis for normalizing overall rates measured over time at steady state and then associating them with what are presumably constant underlying rates of monomer detachment. Different dissolution kinetics models are typically then fit to the data to understand the dissolution behavior¹⁹. Except for limited cases where the micro to nanoscale topology can be monitored during dissolution using techniques like in situ liquid phase AFM^{15,20}, the evolution of topology is typically not directly accessible during experiment. The kinetics is further confounded by the fact that local interfacial curvature can alter dissolution rates through the effect of Laplace pressure on solubility, as described by the Gibbs-Thomson equation. This effect is easily modeled for simple geometries (such as spherical particles), where it makes well-known predictions about the enhanced solubility of small particles, However, the dissolution of hierarchically structured materials is

more difficult to treat. Thus, the combined influence of reactive surface area and particle geometry have historically been difficult to disentangle.

In recent years, a new paradigm of crystal growth has emerged, wherein classically-assumed processes of atom-by-atom or ion-by-ion addition have been supplemented by mechanisms of crystallization by particle attachment (CPA)²¹⁻²³. CPA, particularly a sub-class known as oriented attachment occurs when aggregating particles either fuse to form single crystals, or generate assemblies of aligned but distinct particles known as mesocrystals in which the particles are separated by mostly organics²⁴ or hydration layers²³. Because the crystallinity of the primary particles biases the modes of attachment, the resulting structures are often distinct from those of traditional aggregates, such as the fractal aggregates that are commonly formed by diffusion limited and reaction limited aggregation²⁵. Examples include porous hematite spindles^{26,27}, one dimensional chains and two dimensional superlattices of semiconductor nanoparticles^{28,29}, branched chains of anatase particles³⁰ and boehmite particles³¹, goethite nanorods with extensive internal defects³², and even assemblies of peptides³³ and proteins³⁴.

Page 2: “Many methods have been used to explore dissolution kinetics and pathways, but very few techniques can directly image the dissolution process at a nanometer scale in real time. In situ liquid phase scanning transmission electron microscopy, (s)TEM, is an emerging technique that has shown tremendous progress in revealing pathways and exploring the kinetics of nucleation, crystal growth, assembly, and dissolution. Using in situ liquid phase TEM, we directly observed the dissolution of those two types of Hm structures.”

Page 7: “To understand the roles of particle misorientation and interfacial curvature, we developed a general model for dissolution that accounts for the impact of surface energy, γ_s , grain boundary energy, γ_{GB} , and particle shape. This model (explained in detail in the Supplemental Materials) considers how interfaces (including the particle surface area, S_s , and grain boundary area, S_{GB}) produce a Laplace pressure that changes the thermodynamic driving force for dissolution. Our model calculates the Laplace pressure in terms of generalized coordinates, and thus allows us to easily treat the dissolution of arbitrary shapes. Here, we model necked particles as a series of conical segments, defined by the radii, r_i , where each segment meets. In brief summary, the Laplace pressure associated with a coordinate, q_i , is obtained from the ratio of the derivative in interfacial energy with respect to that coordinate ($\gamma\partial S/\partial q_i$) to the derivative in volume ($\partial V/\partial q_i$). The Gibbs-Thomson equation then allows us to convert the Laplace pressure into changes in local solubility about each segment. Increased solubilities are modeled as an enhancement in the local ion detachment rate (per unit area), which provides the basis for a kinetic model that predicts the evolution of particle shape.”

Page 7: “The solubilities predicted by our model agree with standard implementations of the Gibbs-Thomson equation, which calculate the Laplace pressure in terms of local Gaussian curvature. Simple spherical and cylindrical particles have a positive surface curvature that leads to enhanced solubility for smaller particles. Thus, smaller spherical particles dissolve at an enhanced rate (Fig. 4a,b). However, the necks between particles display a combination of positive curvature (which favors deeper incision at the neck) and negative curvature (which favors neck healing). Competition between these effects determines the evolution of neck shape.”

Our models predict that the negative curvature dominates for sharp necks, which typically cause the neck to heal. Thus, most initially-sharp necks will quickly disappear and not contribute significantly to particle detachment. (Fig. 3i,l). For wider necks, the positive curvature becomes dominant. These necks exhibit some initial healing, but ultimately trend towards more rapid dissolution at the neck than at the adjacent

branch, leading to detachment (Fig. 3j,l). However, the neck profile always remains wide and gently curving, and by the time a finally neck separates, the rest of the branch has significantly dissolved too. Thus, this pathway is also unlikely to produce long-lived individual particles. A third scenario is given by a branch with no initial neck but with a high energy grain boundary (Fig. 3k). The grain boundary also contributes to the Laplace pressure, and causes a sharp incision to form that leads to rapid neck dissolution and break-up (Fig. 3k,l).

On page 8: Thus, these three scenarios exhibit distinct dissolution dynamics (Fig. 3l). More complex dynamics could be expected in a real neck configuration where curvature is variable and is accompanied by varying levels of grain boundary energy. These three scenarios and their combinations provide a rationale for why only certain regions of neck break-up are observed *in situ*. The TEM results (Fig. 2a and b) could be explained by a branch that contains no initial neck in the beginning (0.0s), but where a high-energy grain boundary causes the necking to initiate, and gradually form a wider and deeper neck (Fig. 2a, 4.8s and Fig. 2b 9.6s, indicated by the arrow). Eventually, the rapid dissolution at the neck leads to the break-up and cause the particles to detach.”

Page 9: “The above findings reveal the previously unknown dissolution pathway of DPD, which accelerates dissolution due to the small size of the constituent particles and the strain produced at the interparticle boundaries by the CPA process, which is recognized to occur widely in materials systems as diverse as semiconductors,^{28,29,49,50} metals,^{40,41} silicates,^{51,52} oxides,^{22,23,27,32,53-55} fluorides,⁵⁶ carbonates,⁵⁴ organic compounds,⁵⁷ peptides,³³ and proteins.³⁴ Hence, even though classical monomer-by-monomer dissolution will always be present and, indeed, is still the fundamental process that enables DPD to take place, the nanoscale structure of mesocrystals in this wide range of materials fundamentally alters their dissolution dynamics from that of monolithic single crystals and is thus a direct consequence of growth via CPA. We fully expect these dramatic differences in dissolution behavior to be manifest in other properties. For example, a particle aggregate comprising a dendritic arrangement of rod-like particle chains can be expected to deform through bending of the arms and slip at the interparticle boundaries, possibly leading to a stress-strain relationship resembling plastic deformation instead of that for a brittle single crystal.”

Page 9: “Potential applications that are impacted by either the realization that DPD can dominate dissolution processes or by the opportunities that DPD offers to manipulate dissolution processes when materials are formed via CPA fall into two categories: 1) using the phenomenon of DPD to interpret observations in geological settings and materials processing systems and 2) utilizing the phenomenon to tune dissolution rates in technological settings. In the first case, understanding that minerals formed by CPA — and, thus, those that exhibit a hierarchical morphology — will dissolve at a higher rate, release nanoparticles as they dissolve, and do so in a way that preferentially maintains regions that have undergone the most coarsening, will be critical to interpreting downstream outcomes. Examples include rates and resulting microstructures associated with mineral weathering and/or recrystallization in geochemical reservoirs, riparian environments, and in industrial settings. The latter include processing of aluminum ores into suspensions of aluminum hydroxide particles during the Bayer process⁶⁰ and the processing of legacy nuclear wastes, which are characterized by extensive aggregation of primary nanoparticles^{31,61}. In the second case, as discussed above, creating conditions in which target products form via CPA and using methods to control the primary particle size and strain would establish an ability to tune the rates and products of dissolution, nanoparticle release, and recrystallization with applications in areas like corrosion engineering, nutrient release, and drug delivery.”

Aggregation/agglomeration theories opposing monomer-by-monomer crystallization exist since Smoluchowski [1]. This pioneering work gave rise to the field of population balance models (PBM) [2]. Aggregation/agglomeration (or CPA) tends to generate fractal-type crystals. Including limitation of

particles to be aggregated to the main crystal because of diffusion, which is the basic process of forming fractals, leads to the theory of diffusion limited aggregation (DLA) [3]. Dissolution has already been included in DLA-frameworks [4]. Furthermore, there exists evidence for the detachment of clusters during local dissolution [5]. Finally, there appears to be a relationship to the "chunk effect" discussed in this paper [6], even though research in this area goes back to the 1960s. As these examples demonstrate, there is much prior research, all of which is ignored. In light of these and related works, I would expect deeper insight than a mere observation of the DPD phenomenon that expands our scientific understand further.

[1] M. von Smoluchowski, Versuch einer mathematischen Theorie der Koagulationskinetik kolloider Lösungen, Z. für Phys. Chemie, 92, 129, (1917)

[2] D. Ramkrishna, Population Balances: Theory and Applications to Particulate Systems in Engineering, 1st ed., San Diego (2000).

[3] T.A. Witten, L.M. Sander, Diffusion-Limited Aggregation, a Kinetic Critical Phenomenon, Phys. Rev. Lett. 47, 1400 (1981)

[4] Z. Xu, P. Meakin, Phase-field modeling of two-dimensional solute precipitation/dissolution: Solid fingers and diffusion-limited precipitation, J. Chem. Phys. 134, 044137 (2011)

[5] R.G. Buchheit et al Local Dissolution Phenomena Associated with S Phase (Al₂CuMg) Particles in Aluminum Alloy 2024-T3 J. Electrochem. Soc. 144, 2621 (1999)

[6] F.A. Silveira, F.D.A. Aarão Reis, Detachment of non-dissolved clusters and surface roughening in solid dissolution, Electrochim. Acta 111, 1 (2013)

Reply: We appreciate the review's point that particle aggregation models have existed for many years, and that Smoluchowski's pioneering work led to a quantitative picture. We also agree that the history associated with this work should be addressed in the introduction, as should the examples of cases where particle disaggregation has been reported or implicated, in order to provide proper historical context. With respect to the Smoluchowski analysis and subsequent models of DLA, such aggregation models do not anticipate the mechanism of oriented attachment, which characterizes the class of CPA processes involving crystalline primary particles as the higher order species that drives crystal growth. As has been demonstrated in a range of investigations over the past decade, there are attractive forces, torques and barriers in play that lead to crystallographic coalignment of the particles to produce mesocrystals with a range of architectures that are distinctly different from random fractal aggregates associated with DLA. The figure below, which comes from a Chemical Reviews article currently in press, (Banuelos et al., "Oxide- and Silicate-Water Interfaces and Their Roles in Technology and the Environment" *Chem. Rev.* (In Press) DOI: <https://doi.org/10.1021/acs.chemrev.2c00130>) shows a number of examples (panels A, B, E and F) that differ greatly in architecture from each other and from random fractals generated by DLA, but that all arise from CPA. This is the class of materials addressed by this manuscript.

Figure 58. Hierarchical crystallization pathways lead to novel architectures. (A,B) PbSe nanowire (A)¹⁰²¹ and honeycomb structure sheet (B)¹⁰²² formed by particle assembly. (A) Reproduced from ref ¹⁰²¹. Copyright 2005 American Chemical Society. (B) Adapted with permission from ref ¹⁰²². Copyright 2014 American Association for the Advancement of Science. (C) Calcite pillars growing epitaxially on calcite rhomb via dense liquid phase (DLP).¹⁰²³ (D) Aragonite bundles forming from CaCO₃ DLP droplet. (D) Adapted with permission from ref ¹⁰¹⁷. Copyright 2014 American Association for the Advancement of Science. (E) Spindle-shaped mesocrystal formed by interface-drive nucleation and subsequent particle attachment. (E) Adapted with permission from ref ⁹¹⁹. Copyright 2021 The Authors, under exclusive license to Springer Nature Limited. (F) Branched rutile nanowire where branches start by attachment and transformation of anatase nanoparticles. (F) Adapted from ref ⁸⁷⁷. Copyright 2013 American Chemical Society.

*Reprinted with permission from Banuelos et al., Oxide- and Silicate-Water Interfaces and Their Roles in Technology and the Environment, *Chem. Rev.* 123, 6413–6544 (2023), American Chemical Society.

With regards to the previous cases of particle disaggregation, because of the crystalline nature of mesocrystalline solids is distinct from that of polycrystals or fractal aggregates formed by DLA— i.e., the lattice is continuous or nearly continuous through the interfaces between the primary particles — the dissolution process is not a simple extension of either disintegration of DLA aggregates or the chunk effect. Moreover, even if crystal dissolution were to occur by release of ions or molecular clusters, as opposed to

monomers, the rapid dissolution of mesocrystals generated through CPA, which is due to neck curvature and interfacial strain, would still occur and be characteristic of such crystals, which are ubiquitous across materials classes, as noted in the previous response. Consequently, while the previous examples provide important context, they do not anticipate DPD of CPA-generated mesocrystals, nor do they provide a physical model for interpreting or predicting the progression of DPD or the rate at which it proceeds.

We also want to note that a number of the references cited by the reviewers do not relate to DPD or do so only tangentially.

1. Reference 4 is about phase-field modeling of solute precipitation/dissolution. It draws an analogy to the DLA by forming similar fractal patterns, but that work does not address either aggregation or disaggregation of particles
2. To the best of our knowledge, the literature on particle aggregation related to DLA or RLA, including references 2 and 3, does not address particle disaggregation and we could not find any references to these processes that do so.
3. The “chunk effect” (reference 6) used in the electrochemical community is reported for systems involving bulk polycrystalline metal alloys under applied electric fields. This is arguably a very different scenario than dissolution of mesocrystals in mild solution conditions where the only driving force is thermally activated detachment of ions. However, we completely agree that the “chunk effect” is an excellent example of the universality of this particle detachment phenomenon across many different fields, and we fully expect follow-on studies to be performed in a number of fields including material science, geochemistry and chemical engineering.

To address this comment, in addition to the new text detailed in response to the above comment, we have added significant context and references in the introduction of the revised version: “Although research on CPA has drawn significant interest in recent years²¹, whether, by analogy, the reverse process of dissolution by particle detachment (DPD) occurs and what determines its contribution to the overall dissolution rate relative to concomitant classical monomer-by-monomer dissolution has received little attention. One rare example is the “chunk effect” that has been observed in electrochemical corrosion^{3,35-38}, wherein the disintegration of polycrystalline materials into individual particles is believed to explain anomalously high dissolution rates. However, the polycrystals do not arise from CPA and an electric field is usually involved to generate the effect. None the less, past observations that OA-generated mesocrystals in which the primary particles are separated by intervening hydration layers can disintegrate into the primary particles during sublimation of the hydration layers²³ provides evidence that mesocrystalline solids grown via CPA can undergo DPD. However, whether or not a similar process is possible when the constituent particles have fused to form a continuous crystal lattice has not been explored. Understanding the mechanisms of DPD will be important to modeling the durability and reactivity of diverse materials, especially for the complex hierarchically structured materials that are frequently synthesized using CPA.”

Further comments and questions:

1) The manuscript jumps immediately to the results after the abstract/introduction paragraph. In light of existing prior work, only some of which is discussed above, an extensive introduction presenting the state of the art in chemical dissolution is a necessity.

Reply: We agree with the reviewer that the manuscript will be improved with the additional of more historical perspective. The changes we have made to the introduction to address this are given in the reply to the previous comment. We do want to emphasize again however, that nothing in the previous literature covers the dissolution of mesocrystals formed by crystallization by particle attachment and none of the

studies anticipate the impact of the morphology or defect structure that comes about through this growth mechanism on the enhanced rate of dissolution. We have been careful to point out the differences in the revised introduction.

2) All theory has been moved to SI and only the experimental evidence remains in the main text. This is unfortunate, as a theoretical discussion would give the manuscript more substance. As long as the theory is not in the main text, the manuscript is merely a discussion of the experimental findings. I recommend moving some of the theory into the main text.

Reply: We have followed the reviewers' suggestion and added a significantly more detailed discussion of the dissolution model to page 7 of the main text (primarily by moving content from the supplemental material). We believe that the revised presentation provides adequate details about our modeling approach to enable the reader to understand the ways that competition between surface energy, grain boundary energy, and particle surface area define the tendency for hierarchal structures to dissolve via particle detachment at a rate that can far exceed dissolution of compact single crystals grown classically. The revised reads:

“To understand the roles of **particle** misorientation and **interfacial** curvature, we developed a **general** model for dissolution that accounts for the impact of surface energy, γ_s , grain boundary energy, γ_{GB} , and particle shape. This model (explained in detail in the Supplemental Materials) considers how interfaces (including the particle surface area, S_s , and grain boundary area, S_{GB}) produce a Laplace pressure that changes the thermodynamic driving force for dissolution. Our model calculates the Laplace pressure in terms of generalized coordinates, and thus allows us to easily treat the dissolution of arbitrary shapes. Here, we model necked particles as a series of conical segments, defined by the radii, r_i , where each segment meets. In brief summary, the Laplace pressure associated with a coordinate, q_i , is obtained from the ratio of the derivative in interfacial energy with respect to that coordinate ($\gamma\partial S/\partial q_i$) to the derivative in volume ($\partial V/\partial q_i$). The Gibbs-Thomson equation then allows us to convert the Laplace pressure into changes in local solubility about each segment. Increased solubilities are modeled as an enhancement in the local ion detachment rate (per unit area), which provides the basis for a kinetic model that predicts the evolution of particle shape.

The solubilities predicted by our model agree with standard implementations of the Gibbs-Thomson equation, which calculate the Laplace pressure in terms of local Gaussian curvature. Simple spherical and cylindrical particles have a positive surface curvature that leads to enhanced solubility for smaller particles. Thus, smaller spherical particles dissolve at an enhanced rate (Fig. 4a,b). However, the necks between particles display a combination of positive curvature (which favors deeper incision at the neck) and negative curvature (which favors neck healing). Competition between these effects determines the evolution of neck shape.

Our models predict that the negative curvature dominates for sharp necks, which typically cause the neck to heal. Thus, most initially-sharp necks will quickly disappear and not contribute significantly to particle detachment. (Fig. 3i,l). For wider necks, the positive curvature becomes dominant. These necks exhibit some initial healing, but ultimately trend towards more rapid dissolution at the neck than at the adjacent branch, leading to detachment (Fig. 3j,l). However, the neck profile always remains wide and gently curving, and by the time a finally neck separates, the rest of the branch has significantly dissolved too. Thus, this pathway is also unlikely to produce long-lived individual particles. A third scenario is given by a branch with no initial neck but with a high energy grain boundary (Fig. 3k). The grain boundary also contributes

to the Laplace pressure, and causes a sharp incision to form that leads to rapid neck dissolution and break-up (Fig. 3k,l).

Thus, these three scenarios exhibit distinct dissolution dynamics (Fig. 3l). More complex dynamics could be expected in a real neck configuration where curvature is variable and is accompanied by varying levels of grain boundary energy. These three scenarios and their combinations provide a rationale for why only certain regions of neck break-up are observed *in situ*. The TEM results (Fig. 2a and b) could be explained by a branch that contains no initial neck in the beginning (0.0s), but where a high-energy grain boundary causes the necking to initiate, and gradually form a wider and deeper neck (Fig. 2a, 4.8s and Fig. 2b 9.6s, indicated by the arrow). Eventually, the rapid dissolution at the neck leads to the break-up and cause the particles to detach.

To quantify the dissolution rate at the single particle level, we start by analyzing the time dependence of rhHm particle size, which is fit well by the Gibbs-Thomson equation (Fig. 4a, Movie 1, see SI for methods), thus showing that smaller particles dissolve faster. The spHm consists of primary particles that, individually, are smaller than the rhHm. More importantly, as shown above, necking and particle detachment occur during dissolution, so we expect spHm to dissolve faster than rhHm. To directly compare the rates, we mixed rhHm and spHm to ensure both are subject to same environment (Fig. 4b, see SI for details). The time dependence of the rhHm dissolution rate is again well described by the Gibbs-Thomson relationship (Fig. 4b, blue and green curves). Analyzing the dissolution data for spHm is not as straightforward as the outer particles first dissolve to form the skeleton of rod-like chains of particles before fully dissolving. However, the spindles are almost completely dissolved after ~ 119 seconds. In comparison, the lifetime of a 6 nm spherical particle (which is close to the size of the primary particles that comprise the spHm) is predicted to be ~ 90 seconds (Fig. 4b, orange curve). Thus, the timescale of dissolution for the spindles appears to be defined by the primary particle size, rather than the aggregate size.

3) Lines 57-59: “While the body of each particle dissolves at a more-or-less constant rate (Fig. 2c, open symbols), the necks shrink at an accelerating rate with time”. This acceleration makes sense if one analyses the neck diameter change (L), as dissolution rate is proportional to surface area (L^2). But would that make sense though if one analyses the volume (L^3) of the dissolved crystal? Note that dissolution is a surface-to-volume problem, i.e., the volume of the crystal is dissolved through its surface.

Reply: As the reviewer intuitively, the changes in surface area on growth rate are typically offsetting. Areas with higher surface area can add and subtract atoms more quickly, but the changes are spread across a larger area and thus produce less motion of the interface.

In the model, we define growth/dissolution rates in terms of rates of molecular addition/subtraction per unit area, which can easily be converted to a rate of volume change per unit area, if we consider the molecular volume, v_m . This is expressed as $(dV/dt)/S$. In the experiments, we measure advance/retreat of an interface (which is observed as a change in particle radius). This is expressed as dr/dt .

For most simple geometries (including the ones we study here), the incremental motion of the interface, dr , is given by the incremental change in volume, dV , divided by the surface area. That is, $dr = dV/S$. Consequently, we can show that the two descriptions of dissolution rates discussed above are equivalent:

$$\frac{1}{S} \frac{dV}{dt} = \frac{dr}{dt}$$

Thus, when we observe changes in the rate of interface motion, dr/dt , (such as the accelerating rate of change for the radius as particles shrink), it is a direct reflection of changes in the surface-area normalized rate of molecular detachment across the interface. We hope this interpretation is clearer with the addition of the new discussion of the model to our main text given in the previous response, which includes these statements:

“This model (explained in detail in the Supplemental Materials) considers how interfaces (including the particle surface area, S_s , and grain boundary area, S_{GB}) produce a Laplace pressure that changes the thermodynamic driving force for dissolution.”

“Increased solubilities are modeled as an enhancement in the local ion detachment rate (per unit area)”

4) Lines 59-60: “upon shrinking to a diameter of ~ 1.0 nm, the necks disintegrate”. Could one say this is evidence that the neck cannot sustain anymore the kinetic energy of the system, thus breaking the neck? Notice that the critical value reported for neck fraction is close to 1 nm. How many atomic bonds fit in a distance of 1 nm?

Reply: This is an interesting question. 1 nm is equivalent to 11 atoms across (average Fe-O bonding length is 0.2 nm). We assume that by “kinetic energy” the reviewer is referring to the thermal energy of the system, which is 25 meV at room temperature. This can be compared with the surface free energy gain after neck break-up, which can be calculated as $\gamma * 2\pi r^2$, in which γ is the bulk interfacial energy of the hematite in the system, which is around 0.13 J/m² from our previous calculation (Legg, B. A., Zhu, M., Zhang, H., Waychunas, G., Gilbert, B., & Banfield, J. F. (2016). A model for nucleation when nuclei are nonstoichiometric: Understanding the precipitation of iron oxyhydroxide nanoparticles. *Crystal Growth & Design*, 16(10), 5726-5737.). The calculated energy gain is about 1273 meV, which is nearly 50 times larger than the thermal energy, so it is very unlikely that neck break-up is driven by the thermal energy of the system. To address this comment, we added on page 3:

“Where necks form, we observe that they dissolve more rapidly than the body (Fig. 2c). While the body of each particle dissolves at a more-or-less constant rate (Fig. 2c, open symbols), the necks shrink at an accelerating rate with time (Fig. 2c, closed symbols). Moreover, upon shrinking to a diameter of ~ 1.0 nm, the necks disintegrate, and the particles suddenly detach, thus the neck size jumps to zero (Fig. 2c). **At diameters of under 1nm, the neck only contains a few tens of atoms, and likely becomes difficult to sustain. None the less, we estimate the surface free energy gain to be around 1273 meV during the neck breakup, which is equivalent to changing the neck size from 1 nm to zero, using an interfacial energy of 0.13 J/m² for hematite⁴⁷. This suggest that thermal energy itself, which is only about 25 meV cannot drive neck disintegration.**”

5) Figure 2c: What do the colors represent?

Reply: The colors represent five necking events from supplementary video 5 and 6. We have added this information to the caption and to the excel spreadsheet in the SI. We added,

“The colors represent five necking events from supplementary video 5 and 6.”

6) Lines 92-93: “we then estimate that at pH 7 the mineral surface accessible to the diffusion ranges of e_{aq}^- and O_2^- are within 130 nm and 200 nm, respectively”. What does “mineral surface” mean, and what do these distances represent?

Reply: Here we designed experiments to study the influence of beam induced production of radicals, such as hydrated electrons (e_{aq}^-) and O_2^- on the dissolution. In order to do so, we took snapshots of areas that had not yet been directly exposed to the beam. The “mineral surface” was intended to mean the “surface of the hematite”. We have modified the text to be clearer. “Diffusion length” refers to the “mean diffusion length” calculated by taking the square root of the average lifetime of the radicals multiplied by the diffusion coefficient. Below are the changes to the text.

“We find that the spindles dissolve in the region with no beam irradiation, which implicates the role of diffusion of radicals in the dissolution. In another study²¹, e_{aq}^- and O_2^- were found to be key radiolytic species contributing to akageneite (β -FeOOH) dissolution under the e-beam. If we assume that those species also play a role in Hm dissolution, we then estimate the hematite surface accessible to these beam-generated radicals extends beyond the irradiated region by the characteristic diffusion length of e_{aq}^- and O_2^- , which, at pH 7, are within 130 nm and 200 nm, respectively, for the high dose rate, and 160 nm and 250 nm, respectively, for the low dose rate. The diffusion length here is calculated by taking the square root of the average lifetime of the radicals multiplied by the diffusion coefficient.”

7) Figure 3i-k: I suppose the different lines in the same plots represent different times. Please make the advancement of time clear in the plots.

Reply: We thank the reviewer for pointing this out. To address this, we added arrows to mark the advancement of time in the plot and we added the following to the figure 3 caption: “The contours in i-k indicate the surface profile at evenly spaced increments of time advancing in the direction of the arrows.”

8) Figure 4: “Quantitative analyses of the dissolution and modeling”. Do you mean “of the dissolution model”? In (b), are the red/orange and blue colors swapped? The blue particle appears to be the smallest in (e), while red/orange is the largest. What is the meaning of the colors and squares/circles in (c)? The y-axis title of (c) should be improved. “Fe amount” does not say much.

Reply: We apologize for the confusion. We have changed it to “Quantitative analysis of dissolution based on the Gibbs-Thomson effect”. We have corrected the color. We modified the caption as,

“**Fig. 4 Quantitative analyses of the dissolution based on the Gibbs-Thomson effect.** **a**, Dissolution kinetics of the rhHm fitted by Gibbs-Thomson equation and (**d**) associated snapshots of the dissolution (Movie 1). **b**, Use of the Gibbs-Thomson equation model of different sizes to understand the dissolution of rhHm and spHm. The color in the plot corresponds to the particles highlighted in the images in (e) (Movie 11). Notice that the orange plot corresponds to the 6 nm model. The radius of the spindle is not plotted as

the spindle has a complex morphological evolution. c, Total amount of dissolved Fe versus time for rhHm (open symbols) and spHm (closed symbols) as determined by ICP-OES. The colors represent separate tests on the same condition.”

On page 8, we added.

“To quantify the dissolution rate at the single particle level, we start by analyzing the time dependence of rhHm particle size, which is fit well by the Gibbs-Thomson equation (Fig. 4a, Movie 1, see SI for methods), thus showing that smaller particles dissolve faster. The spHm consists of primary particles that, individually, are smaller than the rhHm. More importantly, as shown above, necking and particle detachment occur during dissolution, so we expect spHm to dissolve faster than rhHm. To directly compare the rates, we mixed rhHm and spHm to ensure both are subject to same environment (Fig. 4b, see SI for details). The time dependence of the rhHm dissolution rate is again well described by the Gibbs-Thomson relationship (Fig. 4b, blue and green curves). Analyzing the dissolution data for spHm is not as straightforward as the outer particles first dissolve to form the skeleton of rod-like chains of particles before fully dissolving. However, the spindles are almost completely dissolved after ~ 119 seconds. In comparison, the lifetime of a 6 nm spherical particle (which is close to the size of the primary particles that comprise the spHm) is predicted to be ~ 90 seconds (Fig. 4b, orange curve). Thus, the timescale of dissolution for the spindles appears to be defined by the primary particle size, rather than the aggregate size.”

9) Minor: In caption of Figure 1, use capital letter in “difference”.

Reply: We thank the reviewer for pointing out the typo, which is now corrected.

10) Gibbs(-)Thom(p)son, with or without hyphen, with or without “p”? Various spelling variants are used throughout the text.

Reply: We thank the reviewer for pointing out this inconsistency. We have modified it throughout the text to be “Gibbs-Thomson”.

Reviewer #4 (Remarks to the Author):

This manuscript proposed a process of dissolution by particle detachment (DPD), which is an inverse counterpart of crystallization by particle attachment (CPA). Through the in-situ liquid phase TEM technology, the authors directly observed the DPD phenomenon of typical mesocrystals of hematite Fe₂O₃ with spindle-shaped morphology. In short, this work provides promising research directions for DPD applications for various fields, which can be published in Nature Communications by addressing the following issues.

Reply: We are grateful for the reviewer’s recognition that “this work provides promising research directions for DPD applications for various fields”, “which can be published in Nature Communications”. We have added significant discussion and addressed the comments as detailed below.

1. Although the authors verified on the material they used that DPD is the inverse process of CPA, the universality of the DPD phenomenon on the materials formed by CPA still needs to be further explored. The

correlation between the DPD behavior and the structure of small crystals in mesoscopic crystals is an example, while the authors also indicated that the dissolution rates of neck and body in the sharp-neck structures are very close (Fig. 4l). Therefore, some relevant expressions need to be reconsidered.

Reply: We appreciate the reviewer's point about the uncertain universality of DPD. Certainly, the model is universally applicable as it simply brings into play the factors of surface energy, curvature, grain boundary energy, and surface area. However, the extent to which it occurs will depend on the degree to which CPA leads to boundaries with necks and incorporated strain, which are dynamic outcomes of CPA that can change depending on formation rate and atomic mobility, as well as the magnitude of the surface free energy, which is inherent to the material. Moreover, the external factors of temperature of the solution in which the mesocrystals form and the time they spend in solution near saturation at temperature are also important, because they impact the extent to which coarsening increases particle size, reduces curvature, relieves strain, and eliminates concave regions. The effect of temperature and time on these growth features was explored in our previous papers (Zhu et al., *Nature* **590**, 416-422, 2021 and Zhu and De Yoreo. *J. Electrochem. Soc.* **169**, 102510, 2022). In this study, they are implicitly incorporated into the analysis via the terms referred to above. We are uncertain as to what the reviewer intended with respect to reconsidering some of the expressions in order to account for these differences. As noted above, the expressions are general as written. Consequently, we have addressed this comment in the manuscript in two ways. In the presentation of the model on page 7, we expanded the discussion of the model to show the general nature of the expressions used and thus their applicability to any material system and morphology. In the discussion on page 9, we discuss the impact of materials-specific parameters and external factors on variations in dissolution behavior between different materials or mesocrystals of the same material grown under different conditions:

On page 7, we write, "To understand the roles of **particle** misorientation and **interfacial** curvature, we developed a **general model for dissolution** that accounts for the impact of surface energy, γ_s , grain boundary energy, γ_{GB} , and particle shape. This model (explained in detail in the Supplemental Materials) considers how interfaces (including the particle surface area, S_s , and grain boundary area, S_{GB}) produce a Laplace pressure that changes the thermodynamic driving force for dissolution. Our model calculates the Laplace pressure in terms of generalized coordinates, and thus allows us to easily treat the dissolution of arbitrary shapes. Here, we model necked particles as a series of conical segments, defined by the radii, r_i , where each segment meets. In brief, the Laplace pressure associated with a coordinate, q_i , is obtained from the ratio of the derivative in interfacial energy with respect to that coordinate ($\gamma \partial S / \partial q_i$) to the derivative in volume ($\partial V / \partial q_i$). The Gibbs-Thomson equation then allows us to convert the Laplace pressure into changes in local solubility about each segment. Increased solubilities are modeled as an enhancement in the local ion detachment rate (per unit area), which provides the basis for a kinetic model that predicts the evolution of particle shape.

The solubilities predicted by our model agree with standard implementations of the Gibbs-Thomson equation, which calculate the Laplace pressure in terms of local Gaussian curvature. Simple spherical and cylindrical particles have a positive surface curvature that leads to enhanced solubility for smaller particles. Thus, smaller spherical particles dissolve at an enhanced rate (Fig. 4a,b). However, the necks between particles display a combination of positive curvature (which favors deeper incision at the neck) and negative curvature (which favors neck healing). Competition between these effects determines the evolution of neck shape.

Our models predict that the negative curvature dominates for sharp necks, which typically cause the neck to heal. Thus, most initially-sharp necks will quickly disappear and not contribute significantly to particle detachment. (Fig. 3i,l). For wider necks, the positive curvature becomes dominant. These necks exhibit some initial healing, but ultimately trend towards more rapid dissolution at the neck than at the adjacent

branch, leading to detachment (Fig. 3j,l). However, the neck profile always remains wide and gently curving, and by the time a finally neck separates, the rest of the branch has significantly dissolved too. Thus, this pathway is also unlikely to produce long-lived individual particles. A third scenario is given by a branch with no initial neck but with a high energy grain boundary (Fig. 3k). The grain boundary also contributes to the Laplace pressure, and causes a sharp incision to form that leads to rapid neck dissolution and break-up (Fig. 3k,l).”

On page 9, after the paragraph that begins with, “The above findings reveal the previously unknown dissolution pathway of DPD, which accelerates dissolution due to the small size of the constituent particles and the strain produced at the interparticle boundaries by the CPA process,” we added,

“The extent to which rates of DPD and resultant dissolution morphologies differ from those seen in compact crystals grown by the classical process of ions or ion clusters addition to steps on faceted crystal surfaces depends on inherent materials parameters, such as solubility, surface tension, and elastic constants (via the grain boundary energy), as well as external factors of temperature and incubation time. While the expressions used in the model presented here are general and include the inherent materials parameters explicitly, the external factors are only implicitly incorporated into the analysis through the as-observed morphology. Higher temperatures and/or longer times spent in solution near the saturation point lead to more extensive coarsening, which increases average particle size, reduces curvature, relieves misalignment and corresponding strain at the boundaries, and eliminates concave regions^{27,58,59} to an extent that also depends on the atomic mobility of the material. The implication is that the extent to which DPD dominates the dissolution process for a given material system cannot be predicted simply based on materials parameters alone; rather, either priori knowledge of the CPA dynamics that led to formation of the mesocrystals or observational data on the resulting morphology and distribution of grain boundaries is required. Conversely, the influence of both internal and external factors implies an ability to tune the speed of the dissolution using methods to control the primary particle size and strain, which could be applied in areas like corrosion engineering, nutrient release, and drug delivery. Future work should explore the factors that define the grain-boundary energies between particles in mesocrystals and how this influences the degree to which DPD occurs.”

2. *Why the dissolution rates of rhHm and spHm measured by ICP-OES are fast at the dissolution beginning? More explanations are needed.*

Reply: We thank the reviewer for pointing out that from our bulk ICP-OES measurement, the dissolution starts fast at the beginning and then slows down. This could be explained by less and less surface area during dissolution and the system is stirred vigorously so diffusion is not considered here. The mass dissolved rate, dM/dt , is given by $4\pi\rho r^2 dr/dt$, in which ρ is the density, M is the dissolved mass at dt , and r is the radius of the particles. The dr/dt should be proportional to $1/r$ from Gibbs-Thomson effect, so the dissolution rate dM/dt is proportional to r . The ratio of dissolution rates at timepoints $t = t_2$ and $t = t_1$ is given by r_2/r_1 , in which t_2 is later than t_1 and r_2 is smaller than r_1 ; therefore, the dissolution gets slower and slower.

We added more discussions as highlighted below.

“To compare the macroscopic dissolution rates of spHm and rhHm, we measured their temporal dissolution profiles using inductively coupled plasma-optical emission spectrometry (ICP-OES) (Fig. 4c). The dissolution of spHm and rhHm are both fast in the beginning and then level off, but the dissolution of spHm is much faster (~7.5 times) than that of rhHm, in agreement with the faster dissolution rate for spHm versus

rhHm at the single particle level. The reduction in dissolution rate with time for both cases can be explained by reducing surface area during dissolution. The mass dissolved rate, dM/dt , is given by $4\pi\rho r^2 dr/dt$, in which ρ is the density, M is the dissolved mass at dt , and r is the radius of the particles. The magnitude of dr/dt should be proportional to $1/r$ due to the Gibbs-Thomson effect, so the dissolution rate dM/dt is proportional to r , and therefore the dissolution gets slower.

3. *Observing a nanostructural transformation process may not be a research goal, but what needs to be known is how to connect the DPD phenomenon with its potential applications. Related discussion and outlook should be included to this work.*

Reply: We agree that additional text to explain how DPD can be connected to potential applications would be a useful addition to the manuscript. We have added the following to the discussion section on page 9 following the paragraph that begins with, “The extent to which rates of DPD and resultant dissolution morphologies differ from those seen in compact crystals grown by the classical process of ions or ion clusters addition to steps on faceted crystal surfaces depends on inherent materials parameters, such as solubility, surface tension, and elastic constants (via the grain boundary energy), as well as external factors of temperature and incubation time” we have added:

“Potential applications that are impacted by either the realization that DPD can dominate dissolution processes or by the opportunities that DPD offers to manipulate dissolution processes when materials are formed via CPA fall into two categories: 1) using the phenomenon of DPD to interpret observations in geological settings and materials processing systems and 2) utilizing the phenomenon to tune dissolution rates in technological settings. In the first case, understanding that minerals formed by CPA — and, thus, those that exhibit a hierarchical morphology — will dissolve at a higher rate, release nanoparticles as they dissolve, and do so in a way that preferentially maintains regions that have undergone the most coarsening, will be critical to interpreting downstream outcomes. Examples include rates and resulting microstructures associated with mineral weathering and/or recrystallization in geochemical reservoirs, riparian environments, and in industrial settings. The latter include processing of aluminum ores into suspensions of aluminum hydroxide particles during the Bayer process⁶⁰ and the processing of legacy nuclear wastes, which are characterized by extensive aggregation of primary nanoparticles^{31,61}. In the second case, as discussed above, creating conditions in which target products form via CPA and using methods to control the primary particle size and strain would establish an ability to tune the rates and products of dissolution, nanoparticle release, and recrystallization with applications in areas like corrosion engineering, nutrient release, and drug delivery.”

4. *Whether the DPD phenomenon is universal, if it is only a very unique system, then this phenomenon may lose its great significance of research. Can we recklessly speculate that this phenomenon may not exist in a system with excellent crystallinity? Besides the unique structure of Fe₂O₃, is it possible to observe it in other systems?*

Reply: We appreciate the reviewer’s question on looking beyond the current system. As discussed above, certainly the physical model and analytical expressions are general. The DPD phenomenon itself is very likely be widely observable in systems beyond hematite and, more broadly, beyond iron oxides. Firstly, there are similar mesocrystal structures in many material and mineral systems as documented in Cölfen, Helmut, and Markus Antonietti. Mesocrystals and nonclassical crystallization. John Wiley & Sons, 2008 and in De Yoreo, James J., et al. "Crystallization by particle attachment in synthetic, biogenic, and geologic environments." Science 349.6247 (2015): aaa6760. As shown in Cölfen and Antonietti’s book, a number

of them form spindle-shaped mesocrystals and, as discussed in the review by De Yoreo et al., particulate boundaries with necks and defects are common features of crystals grown by CPA, which has been observed in oxides, metals, non-oxide semiconductors, phosphates, carbonates, and even polymer, peptide, and protein crystals. While the state of strain in the organic systems is likely to be low, simply because the elastic constants are small, there is no reason to expect the inorganic systems to behave differently. Thus, we fully expect that DPD occur in a wide range of systems under the appropriate conditions. Second, our modeling work has shown that necking and subsequent particle detachment can occur for a significant range of strain energies and values of curvature. While the ranges of values will certainly vary between different material systems, we should still expect to see DPD processes in other systems. Conversely, in systems with excellent crystallinity, such as the compact rhombohedral hematite single crystals studied here, classic dissolution should be observed instead.

We have now included more discussion of existing literature related to DPD. One study pointed out to us by another reviewer, who suggests that DPD may occur in the case of mesocrystal recrystallization before the particles are fully crystallographically fused, is from Brunner, Julian, et al. "Nonclassical recrystallization." *Chemistry—A European Journal* 26.66 (2020): 15242-15248. Another is the so-called "chunk effect", in which detachment of clusters occurs during electrochemical corrosion, leading to a higher dissolution rate compared to the rate calculated from the current (Marsh, G. A., and E. Schaschl. "The difference effect and the chunk effect." *Journal of the Electrochemical Society* 107.12 (1960): 960). This effect usually happens in metallic alloy systems that have grain boundaries and phase boundaries, which can be explained by our model. We believe our in-situ data, modeling work, and the relevant literature indicate that the DPD phenomenon is common across many material systems.

Below are the changes we have made to address this comment:

On page 2, "Although research on CPA has drawn significant interest in recent years²¹, whether, by analogy, the reverse process of dissolution by particle detachment (DPD) occurs and what determines its contribution to the overall dissolution rate relative to concomitant classical monomer-by-monomer dissolution has received little attention. One rare example is the "chunk effect" that has been observed in electrochemical corrosion^{3,35-38}, wherein the disintegration of polycrystalline materials into individual particles is believed to explain anomalously high dissolution rates. However, the polycrystals do not arise from CPA and an electric field is generally involved to generate the effect. None the less, past observations that OA-generated mesocrystals in which the primary particles are separated by intervening hydration layers can disintegrate into the primary particles during sublimation of the hydration layers²³ provides evidence that mesocrystalline solids grown via CPA can undergo DPD. However, whether or not a similar process is possible when the constituent particles have fused to form a continuous crystal lattice has not been explored. Understanding the mechanisms of DPD will be important to modeling the durability and reactivity of diverse materials, especially for the complex hierarchically structured materials that are frequently synthesized using CPA."

On page 6, "... there is a range of orientations between the primary particles, with the mismatch between adjacent particles ranging from about 0° to 10° for the cases investigated here (Fig. 3e-g). As mentioned previously, the continuous nature of the crystal lattice is apparent; this misorientation between grains appears to be facilitated by dislocations, as for a low-angle grain boundary (Fig. 3f). The grain boundary — i.e., strain — energy at these contacts can be estimated from the Shockley-Read equation, which dictates that it increases linearly with respect to the degree of grain mismatch below a certain angle²². The greater recalcitrance to dissolution by the interior skeleton suggests that this region has coarsened to reduce the defect density, coarsen the particles, and eliminate concavities in the branches.

On page 9, “The extent to which rates of DPD and resultant dissolution morphologies differ from those seen in compact crystals grown by the classical process of ions or ion clusters addition to steps on faceted crystal surfaces depends on inherent materials parameters, such as solubility, surface tension, and elastic constants (via the grain boundary energy), as well as external factors of temperature and incubation time. While the expressions used in the model presented here are general and include the inherent materials parameters explicitly, the external factors are only implicitly incorporated into the analysis through the as-observed morphology. Higher temperatures and/or longer times spent in solution near the saturation point lead to more extensive coarsening, which increases average particle size, reduces curvature, relieves misalignment and corresponding strain at the boundaries, and eliminates concave regions^{27,58,59} to an extent that also depends on the atomic mobility of the material. The implication is that the extent to which DPD dominates the dissolution process for a given material system cannot be predicted simply based on materials parameters alone; rather, either priori knowledge of the CPA dynamics that led to formation of the mesocrystals or observational data on the resulting morphology and distribution of grain boundaries is required. Conversely, the influence of both internal and external factors implies an ability to tune the speed of the dissolution using methods to control the primary particle size and strain, which could be applied in areas like corrosion engineering, nutrient release, and drug delivery. Future work should explore the factors that define the grain-boundary energies between particles in mesocrystals and how this influences the degree to which DPD occurs.”

Reviewer #5 (Remarks to the Author):

This study proposes that the inverse of crystallization by particle attachment (CPA, also commonly referred to as “non-classical crystallization”) is dissolution by particle detachment (DPD). The authors provide a demonstration of this phenomenon by examining and comparing the dissolution behavior of hematite nanocrystals with two distinct morphologies (rhombohedral vs. spindle-shaped mesocrystals). They demonstrate that the mesocrystal morphology (a consequence of growth via CPA) exhibits significantly higher rates of dissolution compared with the compact rhombohedral nanocrystals. They suggest that crystals that form via CPA processes may dissolve via DPD due to distinct characteristics of their nanostructure. Specifically, in this case they attribute the enhanced dissolution to the localized impacts of lattice strain and particle curvature, both of which were consequences of the (near) oriented attachment of nanosized precursors that produced branch-like structures.

The work is of excellent quality, is significant in advancing understanding of mineral dissolution, and is novel. They effectively use advanced in situ (and ex situ) imaging methods combined with other techniques to link nanoscale characteristics to macroscopic properties.

I don't see any need for major revisions and support publishing with only minor changes. One aspect that I think should be emphasized (or made a bit clearer) relates to the importance of dissolution by monomer detachment that is occurring in systems involving DPD. In other words, it is generally accepted that classical and non-classical crystallization (I.e., CPA) are not mutually exclusive, even if one dominates the other for a particular material or set of conditions. I think that the authors are demonstrating that this is also the case in the present example (so-called “necking”) during DPD, but would like the text to be a bit more explicit.

Reply: We greatly appreciate the reviewer's recognition of the quality and novelty of the work. We completely agree with the reviewer's point and, in fact, classical dissolution is the process that leads to deepening of shallow necks, more rapid dissolution at grain boundaries, and eventual detachment of particles. We have now added text to clarify that the mechanism of monomer detachment is expected to coexist with DPD in regions away from the interparticle boundaries and plays a central role in the DPD process itself near the boundaries.

On page 2, we have added, "Although research on CPA has drawn significant interest in recent years²¹, whether, by analogy, the reverse process of dissolution by particle detachment (DPD) occurs and what determines its contribution to the overall dissolution rate relative to concomitant classical monomer-by-monomer dissolution has received little attention."

On page 9, we now say, "Hence, even though classical monomer-by-monomer dissolution will always be present and, indeed, is still the fundamental process that enables DPD to take place, the nanoscale structure of mesocrystals fundamentally alters their dissolution dynamics from that of monolithic single crystals and is thus a direct consequence of growth via CPA."

REVIEWERS' COMMENTS

Reviewer #1 (Remarks to the Author):

The authors have done a great job, have thoroughly revised the manuscript, and have solved the multiple comments of the referees. My concerns and comments were completely addressed and as far as I can see also those of the other referees. Therefore, this paper can now be published.

Reviewer #2 (Remarks to the Author):

The paper significantly improved in a major revision and the authors addressed all of our concerns. In particular, the scientific advancement and its impact in the field of crystallization are now clarified in the introduction. The discussions gained with the insertion of parts of the developed model into the manuscript. It is also made clear that the acceleration of dissolution of necks close to rupture is not only due to the surface/volume ratio, which is universal and also occurs in the classic layer-by-layer dissolution, but also due to (i) Laplace pressure (Gibbs-Thomson effect) owed to the curvature in the neck regions leading to an increased driving force of dissolution, and (ii) strain energy accumulated during the formation of the spindles via CPA.

For this reason, the paper is now suitable for publication in Nature Communications after the authors address the following minor comments:

1. The plots need better resolution.
2. Lines 64 and 111: "Nonetheless", in a single word.
3. Line 107: "more or less", no hyphens.
4. Lines 146 to 150: Avoid long phrases. If necessary, use numbering.
5. Fig 3i-k: there is no arrow in the figure.

Reviewer #3 (Remarks to the Author):

The paper significantly improved in a major revision and the authors addressed all of our concerns. In particular, the scientific advancement and its impact in the field of crystallization are now clarified in the introduction. The discussions gained with the insertion of parts of the developed model into the manuscript. It is also made clear that the acceleration of dissolution of necks close to rupture is not only due to the surface/volume ratio, which is universal and also occurs in the classic layer-by-layer dissolution, but also due to (i) Laplace pressure (Gibbs-Thomson effect) owed to the curvature in the neck regions leading to an increased driving force of dissolution, and (ii) strain energy accumulated during the formation of the spindles via CPA.

For this reason, the paper is now suitable for publication in Nature Communications after the authors address the following minor comments:

1. The plots need better resolution.
2. Lines 64 and 111: “Nonetheless”, in a single word.
3. Line 107: “more or less”, no hyphens.
4. Lines 146 to 150: Avoid long phrases. If necessary, use numbering.
5. Fig 3i-k: there is no arrow in the figure.

Reviewer #4 (Remarks to the Author):

The authors have made sufficient revisions and addressed all my concerns. This work can be published now.

Reviewer #5 (Remarks to the Author):

I stand by my comments in the initial review: "The work is of excellent quality, is significant in advancing understanding of mineral dissolution, and is novel. They effectively use advanced in situ (and ex situ) imaging methods combined with other techniques to link nanoscale characteristics to macroscopic properties."

The manuscript is much improved from the additional details included in the introduction and discussion, as well as the general model for dissolution that accounts for the impact of surface energy, grain boundary energy, and particle shape.

I do not have any general specific suggestions for improvement and support publication with only minor revisions, if needed.

Reviewer #1 (Remarks to the Author):

The authors have done a great job, have thoroughly revised the manuscript, and have solved the multiple comments of the referees. My concerns and comments were completely addressed and as far as I can see also those of the other referees. Therefore, this paper can now be published.

Reply: We thank the reviewer for the compliments and previous suggestions that have enhanced this work.

Reviewer #2 and 3 (Remarks to the Author):

The paper significantly improved in a major revision and the authors addressed all of our concerns. In particular, the scientific advancement and its impact in the field of crystallization are now clarified in the introduction. The discussions gained with the insertion of parts of the developed model into the manuscript. It is also made clear that the acceleration of dissolution of necks close to rupture is not only due to the surface/volume ratio, which is universal and also occurs in the classic layer-by-layer dissolution, but also due to (i) Laplace pressure (Gibbs-Thomson effect) owed to the curvature in the neck regions leading to an increased driving force of dissolution, and (ii) strain energy accumulated during the formation of the spindles via CPA.

For this reason, the paper is now suitable for publication in Nature Communications after the authors address the following minor comments:

1. The plots need better resolution.
2. Lines 64 and 111: “Nonetheless”, in a single word.
3. Line 107: “more or less”, no hyphens.
4. Lines 146 to 150: Avoid long phrases. If necessary, use numbering.
5. Fig 3i-k: there is no arrow in the figure.

Reply: We would like to thank the reviewer again for all the comments on our work, and the support of our work to publish in Nature Communication. Below are changes made to specific comments.

1. The plots need better resolution.

Reply: We have provided plots and figures with higher resolution.

2. Lines 64 and 111: “Nonetheless”, in a single word.

Reply: Changed as suggested.

3. Line 107: “more or less”, no hyphens.

Reply: Changed as suggested.

4. Lines 146 to 150: Avoid long phrases. If necessary, use numbering.

Reply: We have shortened the sentence to read:

“If we assume that those species also play a role in Hm dissolution, we then estimate the Hm surface accessible to these beam-generated radicals extends beyond the irradiated region by the characteristic diffusion length of e_{aq}^- and O_2^- . At a pH of 7, for the high dose rate, the characteristic diffusion length of e_{aq}^- and O_2^- length are 130 nm and 200 nm, respectively, while for the low dose rate, they are 160 nm and 250 nm, respectively.”

5. Fig 3i-k: there is no arrow in the figure.

Reply: Arrow is included in the revised figure with higher resolution.

Reviewer #4 (Remarks to the Author):

The authors have made sufficient revisions and addressed all my concerns. This work can be published now.

Reply: Thank you for the support of our work.

Reviewer #5 (Remarks to the Author):

I stand by my comments in the initial review: "The work is of excellent quality, is significant in advancing understanding of mineral dissolution, and is novel. They effectively use advanced in situ (and ex situ) imaging methods combined with other techniques to link nanoscale characteristics to macroscopic properties."

The manuscript is much improved from the additional details included in the introduction and discussion, as well as the general model for dissolution that accounts for the impact of surface energy, grain boundary energy, and particle shape.

I do not have any general specific suggestions for improvement and support publication with only minor revisions, if needed.

Reply: We again appreciate the support of our work from the reviewer.